# A direct comparison of patient-reported outcomes and experiences in alternative models of maternity care in Queensland, Australia

**Yvette D. Miller** *, **Jessica Tone, Sutapa Talukdar, Elizabeth Martin**

School of Public Health and Social Work, Queensland University of Technology, Kelvin Grove, QLD, Australia

* yvette.miller@qut.edu.au

## Abstract

We aimed to directly compare women's pregnancy to postpartum outcomes and experiences across the major maternity models of care offered in Queensland, Australia. We conducted secondary analyses of self-reported data collected in 2012 from a state-wide sample of women who had recently given birth in Queensland (response rate = 30.4%). Logistic regression was used to estimate the odds of outcomes and experiences associated with three models (GP Shared Care, Public Midwifery Continuity Care, Private Obstetric Care) compared with Standard Public Care, adjusting for relevant maternal characteristics and clinical covariates. Of 2,802 women, 18.2% received Standard Public Care, 21.7% received GP Shared Care, 12.9% received Public Midwifery Continuity Care, and 47.1% received Private Obstetric Care. There were minimal differences for women in GP Shared Care. Women in Public Midwifery Continuity Care were less likely to have a scheduled caesarean and more likely to have an unassisted vaginal birth, experience freedom of mobility during labour and informed consent processes for inducing labour, vaginal examinations, fetal monitoring and receiving Syntocinon to birth their placenta, and report highest quality interpersonal care. They had fewer vaginal examinations, lower odds of perineal trauma requiring sutures and anxiety after birth, shorter postpartum hospital stays, and higher odds of a home postpartum care visit. Women in Private Obstetric Care were more likely to have their labour induced, a scheduled caesarean birth, experience informed consent processes for caesarean, and report highest quality interpersonal care, but less likely to experience unassisted vaginal birth and informed consent for Syntocinon to birth their placenta. There is an urgent need to communicate variations between maternity models across the range of outcome and experiential measures that are important to women; build more rigorous comparative evidence for Private Midwifery Care; and prioritise experiential and out-of-pocket cost comparisons in further research to enable woman-centred informed decision-making.

**Data Availability Statement:** The relevant data is available from http://researchdatafinder.qut.edu.au/individual/n31729 (https://data.researchdatafinder.qut.edu.au/dataset/having-a-baby).

**Funding:** The authors received no specific funding for this work.

**Competing interests:** We have read the journal's policy and the authors of this manuscript have the following competing interests: YM has previously received funding for the development of patient decision aids, including resources for women to choose between models of maternity care, and for establishing a state-wide survey of recent maternity consumers' experience of maternity care across different models of care in Queensland, Australia. The funding bodies for that work had no involvement in the research reported here. EM is employed by a health service at which some participants in this study gave birth. EM was not employed by the health service at the time of data collection (2012) and the health service had no involvement or influence in the analysis of the data for the work reported in this manuscript. JT and ST have no competing interests to declare. This does not alter our adherence to PLOS ONE policies on sharing data and materials

## Introduction

A broad range of maternity model of care (MMC) options are available to Australian maternity care consumers. The options are distinguished by the discipline, continuity of carer and women's capacity to choose the primary care provider, options for planned place of birth, and access to obstetric interventions and procedures [1]. Maternity models of care also differ in how care is funded. Australia has a two-tier system comprising of Medicare, a national publicly funded universal health care program which provides medical care at public hospitals with no out-of-pocket costs to patients, alongside a private health system available through purchasing private health insurance (with some additional out-of-pocket costs). Consumers may also pay the full cost of privately funded care out-of-pocket. Woman-centred maternity service delivery and the wellbeing of women and their babies depends on ensuring every woman receives maternity care appropriate to her needs, values, and circumstances via good quality MMC decision-making and referral in early pregnancy [2]. Providing women with the likelihood of comparable outcomes for each available model is important for informed decision-making regarding MMC [2], which has been shown be to associated with higher patient self-efficacy and satisfaction with the MMC received [3].

At least ten distinct MMCs are offered in Australia that can be broadly distinguished across five categories: Standard Public Care, General Practitioner (GP) Shared Care, Public Midwifery Continuity Care, Private Obstetric Care and Private Midwifery Care (see Table 1) [4–6]. The majority of Australian women (90.4%) are interested in having access to clear and trustworthy information on MMCs [7]. Almost all pregnant women visit a GP as their first care provider for referral to a specific MMC [4]. Limited knowledge about differences between

**Table 1. Maternity model of care categories and definitions and their relationship to the current Maternity Care Classification System (MaCCS) in Australia [5, 18].**

| Model of Care Category | Primary Care Provider | Location of Pregnancy Check-ups | Location of Birth | Who Pays for Care | Relevant Maternity Care Classification System Categories |
|---|---|---|---|---|---|
| Standard Public Care | Rostered hospital midwives and obstetricians | Public hospital or community clinic | Public hospital | Government funded through Medicare (no out-of-pocket costs to women). | Public hospital maternity care; Shared care; Public hospital high-risk maternity care; Remote area care |
| GP Shared Care | Antenatal care: Community maternity service provider (GP/doctor and/or midwife Intrapartum and early postnatal care: Public rostered hospital midwives and obstetricians | GP clinic or public hospital | Public hospital | Antenatal care may incur some out-of-pocket costs for women to pay for GP visits (with some reimbursement from Medicare). Hospital-based care is government funded through Medicare. | Shared care; Combined care |
| Public Midwifery Continuity Care | Primary hospital midwife or small team of hospital midwives (team, caseload/group practice) with obstetric support from rostered hospital obstetricians as required. | Public hospital, community clinic, or birth centre | Public hospital or co-located birth centre | Government funded through Medicare (no out-of-pocket costs to women) | Team midwifery care; Midwifery group practice (caseload) care |
| Private Obstetric Care | A private obstetrician of choice (providing continuity of care) and rostered private hospital midwives | Private hospital or private clinic | Private hospital (where the chosen private obstetrician has visiting rights) | Women's private health insurance (with some out-of-pocket costs for women). Women without private health insurance will incur the full fee (with some Medicare rebates available). | Private obstetrician (specialist) care; General Practitioner obstetrician; Privately practising midwife joint care |
| Private Midwifery Care | A private midwife of choice (providing continuity of care), with option to transfer/incorporate specialist obstetric care from other models that provide it. | Patient's home or midwife's clinic | At home or in a Public Hospital (where the chosen midwife has visiting rights) | Women (Medicare rebates available for antepartum and postpartum care). Public hospital birth costs covered by Medicare. | Private Midwifery Care |

available MMCs makes it difficult for health professionals to support women's informed deci-
sion-making by providing comprehensive and unbiased information on all available options
[4, 5]. In an Australian study, only 7.7% of women were informed of all broad categories of
available MMCs in pregnancy [4]. Women were more likely to be informed about GP Shared
Care and Standard Public Care during their first antenatal visits, and more likely to be
informed of Private Obstetric Care if they had private health insurance [4].

Current evidence comparing MMCs in Australia is largely from randomised controlled tri-
als (RCTs) or cohort studies that compare only two MMCs, primarily using clinical records of
outcomes [5]. Most evidence compares two specific publicly funded MMCs implemented
within a specific birthing facility–Public Midwifery Continuity Care and Standard Public Care
[8–14]. This evidence consistently reveals shorter hospital stays, higher rates of unassisted vagi-
nal births and spontaneous onset of labour, and lower rates of caesarean births, epidurals, and
episiotomies in Public Midwifery Continuity Care compared to Standard Public Care [5].
Consistent with the Australian literature comparing MMCs [5] comparative studies conducted
internationally have primarily focused on comparing midwifery-led care with other models on
clinical outcomes [15–17]. A recent systemic review and meta-analysis of studies comparing
midwifery-led care with standard or usual care found lower odds of caesarean births and episi-
otomy for women in midwifery-led models [17]. An earlier systematic review and meta-analy-
sis comparing midwifery continuity care with other models of care (examined as one
comparison model comprising Obstetrician-led, GP Shared Care, or Standard Care) interna-
tionally indicated that women in midwifery-led models were less likely to have an epidural
during labour, an assisted vaginal birth and an episiotomy and more likely to experience an
unassisted vaginal birth, regardless of their pregnancy risk or if they received caseload versus
team midwifery care [16]. Women in midwifery-led models were also more likely to demon-
strate a higher level of satisfaction with care compared to other models [16]. Despite these dif-
ferences, women are generally not informed about Midwifery Continuity Models of Care by
their GPs, who are responsible for the majority of referrals to a specific MMC [4]. Evidence
that overwhelmingly limits direct comparisons to only two publicly funded models is also
insufficient for meeting the decision-making needs of maternity care consumers who are most
interested in being informed about MMC differences that vary across publicly and privately
funded models, including access to choices in their care provider, place of birth and mode of
birth; after hours provider contact; and continuity of care in labour and birth [7].

Only one study has directly compared privately funded MMCs (Private Obstetric Care)
with publicly funded models (Standard Public Care and a specific (caseload) Public Midwifery
Continuity Care model), using cross-sectional data from only first-time mothers in one spe-
cific hospital [19]. Higher rates of elective caesarean birth, epidurals administered during the
first stage of labour, and episiotomies were found for women who received care in the Private
Obstetric model compared to both Standard Public Care and Public Midwifery Continuity
Care. Compared to the Private Obstetric Model, women who received Public Midwifery Con-
tinuity Care had higher rates of spontaneous labour onset and vaginal birth and lower rates of
analgesia and their baby's admission to neonatal special care units [19]. There is no evidence
about the likelihood of outcomes and/or experiences of all existing distinct maternity models
together for useful direct comparison [5]. A further limitation of extant methods is the com-
parison of only two or three distinct models implemented within one or a small number of
birthing facilities. The substantial variation in the way MMCs are both defined and operatio-
nalised across different facilities and regions limits the generalisability of current evidence to
maternity service delivery in other locations [6, 20].

There remains a paucity of studies comparing the various MMCs available to women
beyond the comparison of only two models for useful informed MMC decision-making at

both an Australian and international level. The available evidence also prioritises direct comparisons between MMCs on the basis of clinical outcomes, ignoring women's other prioritised needs for MMC decision making, such as their likelihood of being supported to make fully informed decisions about medical procedures, and rates of skin-to-skin contact after birth [7]. An international systematic review and meta-analysis found insufficient studies comparing MMCs on the basis of women's experiences of care to warrant their inclusion in synthesising effects [17]. Only four studies comparing MMCs in Australia over the last twenty years have compared women's experiences of care beyond clinical outcomes, and these studies all limited comparisons to between Public Midwifery Continuity Care and Standard Public Care Models [5, 21–24]. A known care provider during labour, better reported quality of interpersonal care, shorter wait times for care visits and mobility during labour were more likely in Public Midwifery Continuity Models than in Standard Public Care [5]. Current evidence therefore lacks not only direct comparisons of all available MMC options in Australia, but generally overlooks comparing the MMC features that are most important to women [4].

In the absence of any prospective trials directly comparing the full range of alternative models of care in Australia, observational data that includes valid assessment of women's MMC can offer some indicator of how women's outcomes and experiences vary. The Maternity Care Classification System (MaCCS) has recently been implemented in Australia to standardise reporting of MMCs and provide opportunities for comparative analyses of maternal and neonatal outcomes using population-level routine clinical reporting [25, 26]. The MaCCS was implemented in Queensland from 2020 [27], but to date, has not been used to report on differences in outcomes between women receiving alternative MMCs on offer. Furthermore, without linkage to supplementary (non-routine) data, information collected using the MaCCS does not provide for comparisons between MMCs beyond clinical outcomes, such as women's self-reported experiences. Population-based survey data that assesses women's self-reported maternity care outcomes and experience is an alternative option to compare MMCs.

## Aim

The aim of this study was to directly compare women's pregnancy, labour, birth, and postpartum outcomes and experiences across the major MMC categories offered in Queensland, Australia, using data collected in 2012 from a state-wide sample of women who had given birth in Queensland. Our comparison sought to estimate the relative likelihood of outcomes and experiences associated with each MMC to support informed MMC decisions. Our intention was to directly compare broad categories of available MMCs in all women, regardless of their risk, with additional adjustment for factors that may result in MMC selection or allocation bias for which we had available data. Methodological decisions prioritised the yielding of comparisons that were optimally useful to women in their MMC decision-making or those seeking to use evidence to support such decisions.

## Methods

### Survey procedure and participants

This retrospective cohort study was conducted using data obtained from the Having a Baby in Queensland Survey, 2012 [28]. Participants were sampled from databases of compulsory birth notification and registration records, held by the Queensland Registry of Births, Deaths and Marriages. All women were eligible to participate if they gave birth in the state of Queensland, Australia between 1st October 2011 and 31st January 2012 and were not found to have had a baby that died more than 28 days after birth or a multiple birth where at least one baby died after 28 days (through cross-checking death records). Eligible women were mailed a survey

package three to four months following birth. All survey packages were addressed and sent by post from the Registry of Birth, Deaths and Marriages to protect confidentiality and anonymity. The survey package included a letter of invitation from the Registrar General of Births, Deaths and Marriages, an English-language information sheet about the study, an English-language paper version of the Having a Baby in Queensland Survey, 2012, participation instructions in 19 other languages (Cantonese, Mandarin, Greek, Korean, Persian, Russian, Serbian, Spanish, Turkish, Vietnamese, German, Arabic, French, Samoan, Filipino, Dinka, Japanese, Khmer and Amharic), and a pen. Women who experienced a multiple birth, a stillbirth or a neonatal death (up to 28 days after birth) were mailed a tailored version of the written survey. Participation in the survey was voluntary. Women were able to (i) complete and return the paper survey booklet using a reply-paid envelope included in the survey package, (ii) complete the survey online, or (iii) complete the survey by telephone with a trained female-identifying interviewer and, if required, a Translating and Interpreting Service interpreter. All women in the live singleton and multiple birth samples, excluding those who experienced a neonatal death, were sent a reminder and thank you postcard two weeks following the mailing of the survey package. The survey and subsequent analyses received clearance from The University of Queensland's Behavioural & Social Sciences Ethical Review Committee (Clearance #2011001083). Participation in the anonymous survey was assessed by the Ethical Review Committee as involving no more than minimal risk of harm and not requiring written consent. Return of the anonymous survey was taken as evidence of consent to participate. The sample used for this analysis includes only women who had a live singleton or multiple birth. Where women had a multiple birth, relevant indicators for the first baby born were used.

## Measures

The *Having a Baby in Queensland Survey, 2012* [28] was a 28-page retrospective population survey of women's self-reported pregnancy, labour, birth, and postpartum experiences and clinical outcomes. Participants were asked to report on their most recent birth (index birth). Survey items relevant to the current analyses are detailed below and in S1 Table.

**Sociodemographic characteristics and reproductive history.** Data on participant's sociodemographic characteristics and reproductive history included age at the time of the index birth, pre-pregnancy body mass index (BMI; derived from self-reported height and weight), area of residence, highest level of education, Aboriginal and Torres Strait Islander identification, language(s) spoken at home, country of birth, parity, and previous caesarean birth. Further details of sociodemographic and reproductive history measures are provided in S1 Table.

**Complications arising during index pregnancy.** A range of complications that can arise during pregnancy were assessed for participants' index pregnancy, including depression, anxiety, gestational diabetes, hypertension/pre-eclampsia, placenta praevia, concerns with the amount of amniotic fluid, problems with the cervix, problems with the baby's cord, baby's size, preterm labour, and ruptured membranes in the absence of labour (see S1 Table for further details).

**Maternity model of care.** Women were classified into one of five different MMCs offered in Queensland: Standard Public Care, GP Shared Care, Public Midwifery Continuity Care, Private Obstetric Care, and Private Midwifery Care (see Table 2) using an SPSS syntax coding algorithm. To assess which MMCs were provided by each facility, an audit of all birth facilities in Queensland were used alongside several items from the *Having a Baby in Queensland Survey, 2012* (e.g., the name of the hospital or birth centre where women had their baby) to create the MMC coding algorithm. Women who reported birthing in a facility that was identified in

**Table 2. Maternity model of care definitions in the current study and their relationship to the current Maternity Care Classification System (MaCCs) in Australia [18].**

| Model of Care Categories in the Current Study[1] | Model of Care Category presented in the Survey | Description of Model of Care Category in the Survey | Relevant Maternity Care Classification System Categories (8) |
|---|---|---|---|
| Standard Public Care | Standard Care in a Public Hospital | Pregnancy check-ups with midwives and/or obstetricians in the public hospital or in a community clinic. Labour and birth in a public hospital. | Public hospital maternity care; Shared care; Public hospital high-risk maternity care; Remote area care |
| GP Shared Care | GP Shared Care | Regular pregnancy check-ups with your GP and some check-ups with midwives and/or obstetricians in the public hospital or in a community clinic. Labour and birth in a public hospital. | Shared care; Combined care |
| Public Midwifery Continuity Care | Midwifery-led Care (Team, Caseload, or Midwifery Group Practice) | Pregnancy check-ups with one midwife or a small team of midwives who work in a public hospital. Labour and birth in a public hospital (with the midwife or midwives that cared for you in pregnancy). | Team midwifery care; Midwifery group Practice (caseload) care |
| | Birth Centre Care | Pregnancy check-ups with one midwife or a small team of midwives who work in a birth centre. Labour and birth in the birth centre. | |
| Private Obstetric Care | Private Obstetric Care | Pregnancy check-ups with a private obstetrician (who you chose). Labour and birth in a private hospital with care provided by your obstetrician and/or hospital midwives. | Private obstetrician (specialist) care; General Practitioner obstetrician; Privately practising midwife joint care |
| Private Midwifery Care | Private Midwifery Care with Birth at Home | Pregnancy check-ups at home with a private midwife (who you chose). Labour and birth at home with care provided by your midwife. | Private Midwifery Care |
| | Private Midwifery Care with Birth in Hospital | Pregnancy check-ups at home with a private midwife (who you chose). Labour and birth in a public hospital (with care provided by your midwife or hospital midwives). | |

[1] Primary care provider, location of pregnancy check-ups and location of birth as described in Table 1.

the audit to only offer one MMC were classified into the MMC that their birth facility offered. Women who birthed in a facility that offered multiple MMCs were classified by both their birth facility and their responses to an open-text question that asked women to identify the type of maternity care they received. To assist women in describing the care they received, a list of seven of the major models of care offered in Australia and their definitions were presented (See column 2, Table 2) and women were asked *'What type of pregnancy and labour/ birth care did you have? Please choose from the list above or describe your experience'*. Two of the seven models of care presented in the *Having a Baby in Queensland Survey* 2012 were combined: Birth centre care (only available through publicly funded services) was combined with Midwifery-led care (team, caseload, or midwifery group practice) to create the Public Midwifery Continuity Care category. Private Midwifery Care with birth at home and Private Midwifery Care with birth in hospital were combined to create the Private Midwifery Care category. Internal validity checks were performed to manually compare classifications determined by the coding algorithm against qualitative data from the primary MMC item and several other useful survey items (*'Why did you have your baby here?'* in reference to birth facility name, *'Were you a private patient or a public patient when you gave birth to your baby?'*, *'Did you have a Private Obstetrician or Private Midwife?'*, and *'Was there one person who coordinated your pregnancy care and provided the majority of your pregnancy check-ups?'*). Cases that were flagged during the internal validity process were manually re-classified. Cases that were unable to be classified by the coding algorithm were manually reviewed and classified into a MMC if sufficient information allowed.

The MMC variable was created prior to the implementation of the MaCCS in Australia. The MaCCS specifies 11 major models of care distinguished by characteristics of the model as

it is intended to be delivered, including patient characteristics, primary care providers, how the care is provided and the location of care [25]. These 11 models can be broadly categorised under the five main models of care defined in the current study based on characteristics of the primary care provider and where and how care is provided (see Table 2) [18].

Classification of MMC did not capture transitions between models of care during pregnancy. Where women transitioned between models of care during the index pregnancy, women were classified into the final model received in pregnancy to allow comparison of outcomes under a single model of care received. Women who reported transitioning between models as a result of an unplanned change of birth location after the onset of labour, a transfer to a hospital from an adjoining birth centre during labour due to complications, or a transition between GP Shared Care during pregnancy and Standard Public Care during labour and birth were classified based on the primary model of care they reported prior to the transition. Some women described their model of care in ways that did not directly distinguish between Public Midwifery Continuity and Standard Public Care models consistent with the description provided in the survey. In such cases, women who birthed in a facility offering both Standard Public and Midwifery-led MMCs were only classified as receiving Public Midwifery Continuity Care if they (a) described care by midwives or a team of midwives only and (b) reported continuity of midwifery-led care, defined by four or less midwives across all stages of care, in an additional survey item. The current analyses include comparisons between four of the five models of care: Standard Public Care, GP Shared Care, Public Midwifery Continuity Care, Private Obstetric Care. Respondents who reported receiving Private Midwifery Care were excluded due to insufficient numbers.

**Experiences and outcomes.** A range of outcomes were compared across the four MMCs in four key domains: (1) obstetric interventions and maternal and infant health outcomes, (2) information provision and decision-making, (3) other maternal experiences of care during pregnancy, labour/birth, and postpartum, and (4) quality of interpersonal care. Further details of outcome measures are provided in S1 Table.

## Data analysis

Data was analysed using IBM SPSS Statistics (version 28). Missing data was excluded listwise across all analyses, unless categorical variables had > 5% of cases missing, where missing cases were retained as a unique level of the variable. One-sample Chi-squared statistics were used to assess equivalence between the study sample and Queensland population sample of women who birthed in 2011 [29, 30] on sociodemographic and reproductive characteristics, where data was available for the Queensland population. Chi-squared statistics were used to assess equivalence between MMCs on sociodemographic characteristics, reproductive history, and complications arising during index pregnancy. Variables found to be non-equivalent at $p < 0.05$ were considered as potential confounders in subsequent analyses. Three series of logistic regression analyses were conducted to estimate the odds of each outcome associated with each MMC, with Standard Public Care as the referent. Unadjusted logistic regression analyses were performed to elicit crude odds ratios (ORs) followed by multivariable logistic regression analyses to adjust for sociodemographic and reproductive history variables that were significantly non-equivalent between MMCs. Third, a final series of multivariable logistic regression models were performed to include additional adjustments for significant complications arising during the index pregnancy and mode of birth and clinical covariates, where clinically applicable. Adjusted models are reported together in one table for each type of outcome measure: obstetric interventions and maternal and infant health outcomes; information provision and decision-making; experiences of care; and quality of interpersonal care. In the final

regression models, parity was considered a clinical covariate of all outcomes. Mode of birth was adjusted for in all models predicting outcomes that co-occur with type of birth: induction of labour; epidural or spinal block for pain relief during labour; mobility during labour; constant fetal monitoring during labour; vaginal examinations during labour; perineal trauma; support people welcome during labour and birth; maternal length of hospital stay; skin-to-skin contact first time holding baby; breastfeeding outcomes; preterm birth; low birthweight; NICU admission; and decision-making for interventions that occur during labour or vaginal birth. Neonate admission to NICU was adjusted for models predicting breastfeeding outcomes. The results of all regression models were considered statistically significant at $p < 0.01$ to reduce type I error associated with multiple comparisons and are presented as crude/adjusted ORs and 99% confidence intervals (CIs). Frequencies for each outcome across each MMC and crude ORs are presented in S2–S5 Tables.

## Results

### Sample

Of the 19,194 eligible women who were mailed surveys, 5,840 (30.4%) responded. From the completed surveys, 2,802 (48.0%) had complete data across the MMC variable, all outcomes, and included confounding variables. The 3,038 excluded participants comprised: 37 women who received care in a Private Midwifery Care model; 204 who could not be classified to a MMC; 266 participants with missing data on the MMC measures; and 2,531 participants who had incomplete data for outcomes and confounding variables with <5% missing. Of the 2,802 women included in the sample for analyses, 510 (18.2%) received Standard Public Care, 609 (21.7%) received GP Shared Care, 362 (12.9%) received Public Midwifery Continuity Care, and 1,321 (47.1%) received Private Obstetric Care. Women who received alternative MMCs were significantly non-equivalent across all sociodemographic and reproductive history variables except for parity, as well as depression, gestational diabetes, concerns with the amount of amniotic fluid, and the baby's size being 'too small' during the index pregnancy (Table 3). Frequencies of multiple births and Aboriginal and Torres Strait Islander women were too low to include as adjustments in subsequent logistic regression models. Relative to the total population of Queensland birthing women in 2011, the study sample marginally overrepresented women from a major city (63.2% in the study sample vs. 61.3% of all women birthing in Queensland, $X^2$ (1) = 4.448, $p$ = .035) and women who were a normal weight (BMI = 18.50–24.99; 53.7% vs. 50.6% of all birthing women, $X^2$ (1) = 10.853, $p$ < .001) (Table 3). Women in the study sample were also more likely to be born in Australia (83.0% vs. 77.0%, $X^2$ (1) = 55.034, $p$ < .001) and primiparous (48.6% vs. 41.1%, $X^2$ (1) = 65.871, $p$ < .001) compared to all birthing women in Queensland. The study sample underrepresented women who lived in a remote or very remote area (2.6% vs. 4.2%, $X^2$ (1) = 17.710, $p$ < .001), younger women (aged < 25; 12.2% vs. 22.0%, $X^2$ (1) 155.503, $p$ < .001) and Aboriginal and Torres Strait Islander women (1.0% vs. 6.0%, $X^2$ (1) = 123.555, $p$ < .001) compared to all birthing women in Queensland.

### Outcomes

**Obstetric intervention and maternal and infant health outcomes.** The odds of receiving an epidural for pain relief during labour, continuous fetal monitoring, having an intact perineum, being diagnosed with depression or anxiety after birth, or a maternal or infant hospital readmission did not differ between Standard Public Care and any other MMC.

After adjusting for non-equivalent sociodemographic and reproductive history variables and clinical covariates, women who received Public Midwifery Continuity Care were more

**Table 3. Sociodemographic, reproductive history, and index birth characteristics by maternity model of care.**

| | Standard Public Care (n = 510) | GP Shared Care (n = 609) | Public Midwifery Continuity Care (n = 362) | Private Obstetric Care (n = 1321) | p | Total Study Sample (n = 2802) | Queensland Population Births 2011 (n = 61125)[1] |
|---|---|---|---|---|---|---|---|
| | n (%) | n (%) | n (%) | n (%) | p | n (%) | n (%) |
| Maternal age at birth | | | | | < .001 | | |
| < 25 | 119 (23.3) | 108 (17.7) | 72 (19.9) | 44 (3.3) | | 343 (12.2) | 13,427 (22.0) |
| 25–29 | 161 (31.6) | 217 (35.6) | 115 (31.8) | 340 (25.7) | | 833 (29.7) | 17,835 (29.2) |
| 30–34 | 139 (27.3) | 179 (29.4) | 119 (32.9) | 558 (42.2) | | 995 (35.5) | 17,688 (28.9) |
| ≥ 35 | 91 (17.8) | 105 (17.2) | 56 (15.5) | 379 (28.7) | | 631 (22.5) | 12,175 (19.9) |
| Pre-pregnancy BMI[2] | | | | | < .001 | | |
| Underweight (<18.5) | 22 (4.3) | 21 (3.4) | 23 (6.4) | 56 (4.2) | | 122 (4.4) | 3,161 (5.3) |
| Normal (18.5–24.99) | 239 (46.9) | 323 (53.0) | 207 (57.2) | 736 (55.7) | | 1505 (53.7) | 30,394 (50.6) |
| Overweight (25–29.99) | 97 (19.0) | 124 (20.4) | 67 (18.5) | 288 (21.8) | | 576 (20.6) | 14,745 (24.6) |
| Obese (≥30) | 104 (20.4) | 96 (15.8) | 37 (10.2) | 172 (13.0) | | 409 (14.6) | 11,758 (19.6) |
| Missing data | 48 (9.4) | 45 (7.4) | 28 (7.7) | 69 (5.2) | | 190 (6.8) | 1,054 (1.7) |
| Area of residence | | | | | < .001 | | |
| Major city | 255 (50.0) | 362 (59.4) | 238 (65.7) | 917 (69.4) | | 1772 (63.2) | 37,134 (61.3) |
| Inner regional | 144 (28.2) | 124 (20.4) | 43 (11.9) | 213 (16.1) | | 524 (18.7) | 11,529 (19.0) |
| Outer regional | 89 (17.5) | 102 (16.7) | 70 (19.3) | 172 (13.0) | | 433 (15.5) | 9,407 (15.5) |
| Remote, very remote | 22 (4.3) | 21 (3.4) | 11 (3.0) | 19 (1.4) | | 73 (2.6) | 2,522 (4.2) |
| Education | | | | | < .001 | | |
| Grade 10 or less | 55 (10.8) | 58 (9.5) | 31 (8.6) | 36 (2.7) | | 180 (6.4) | n/a |
| Vocational education | 169 (33.1) | 248 (40.7) | 124 (34.3) | 297 (22.5) | | 838 (29.9) | n/a |
| Grade 12 or equivalent | 103 (20.2) | 102 (16.7) | 61 (16.9) | 152 (11.5) | | 418 (14.9) | n/a |
| Tertiary Education | 183 (35.9) | 201 (33.0) | 146 (40.3) | 836 (63.3) | | 1366 (48.8) | n/a |
| Aboriginal and/or Torres Strait Islander identification[a] | | | | | .017 | | |
| Aboriginal and/or Torres Strait Islander | 9 (1.8) | 9 (1.5) | 5 (1.4) | 5 (0.4) | | 28 (1.0) | 3,646 (6.0) |
| None | 499 (98.2) | 596 (98.6) | 355 (98.6) | 1313 (99.6) | | 2763 (99.0) | 57,453 (94.0) |
| Language spoken at home | | | | | .006 | | |
| English only | 473 (92.7) | 577 (94.7) | 33 (92.3) | 1268 (96.) | | 2652 (94.6) | n/a |
| Other language(s) | 37 (7.3) | 32 (5.3) | 28 (7.7) | 53 (4.0) | | 150 (5.4) | n/a |
| Country of birth | | | | | < .001 | | |
| Australia | 421 (82.5) | 481 (79.0) | 277 (76.5) | 1148 (86.9) | | 2327 (83.0) | 47,093 (77.0) |
| Other | 89 (17.5) | 128 (21.0) | 85 (23.5) | 173 (13.1) | | 475 (17.0) | 14,032 (23.0) |
| Parity | | | | | .053 | | |
| Primiparous | 225 (44.1) | 293 (48.1) | 193 (53.3) | 652 (49.4) | | 1363 (48.6) | 25,132 (41.1) |
| Multiparous | 285 (55.9) | 316 (51.9) | 169 (46.7) | 669 (50.6) | | 1439 (51.4) | 35,993 (58.9) |
| Previous caesarean birth | | | | | < .001 | | |
| At least one | 74 (14.5) | 84 (13.8) | 22 (6.1) | 271 (20.5) | | 451 (16.1) | 10,711 (17.5) |
| None | 436 (85.5) | 525 (86.2) | 340 (93.9) | 1050 (79.5) | | 2351 (83.9) | 50,414 (82.5) |
| Birth Plurality | | | | | .007 | | |
| Singleton | 500 (98.0) | 605 (99.3) | 362 (100.0) | 1294 (98.0) | | 2761 (98.5) | 60,098 (98.3) |
| Multiple | 10 (2.0) | 4 (0.7) | 0 (0.0) | 27 (2.0) | | 41 (1.5) | 1,025 (1.6) |

*(Continued)*

**Table 3.** (Continued)

| | Standard Public Care (n = 510) | GP Shared Care (n = 609) | Public Midwifery Continuity Care (n = 362) | Private Obstetric Care (n = 1321) | p | Total Study Sample (n = 2802) | Queensland Population Births 2011 (n = 61125)[1] |
|---|---|---|---|---|---|---|---|
| | n (%) | n (%) | n (%) | n (%) | p | n (%) | n (%) |
| Complications during index pregnancy | | | | | | | |
| Depression | 27 (5.4) | 38 (6.2) | 12 (3.3) | 35 (2.6) | < .001 | 112 (4.0) | n/a |
| Anxiety | 62 (12.2) | 69 (11.3) | 32 (8.8) | 127 (9.6) | .252 | 290 (10.3) | n/a |
| Gestational diabetes | 56 (11.0) | 41 (6.7) | 13 (3.6) | 104 (7.9) | < .001 | 214 (7.6) | n/a |
| Hypertension/pre-eclampsia | 59 (11.6) | 62 (10.2) | 33 (9.1) | 140 (10.6) | .865 | 294 (10.5) | n/a |
| Placenta praevia | 40 (7.8) | 50 (8.2) | 30 (8.3) | 100 (7.6) | .547 | 220 (7.9) | n/a |
| Problem with cervix | 14 (2.7) | 10 (1.6) | 5 (1.4) | 35 (2.6) | .533 | 64 (2.3) | n/a |
| Amount of amniotic fluid was a concern | 33 (6.5) | 23 (3.8) | 16 (4.4) | 35 (2.6) | .002 | 107 (3.8) | n/a |
| Problem with baby's cord | 11 (2.2) | 9 (1.5) | 2 (0.6) | 31 (2.3) | .425 | 53 (1.9) | n/a |
| Baby was too big | 37 (7.3) | 34 (5.6) | 13 (3.6) | 74 (5.6) | .169 | 158 (5.6) | n/a |
| Baby was too small | 56 (11.0) | 43 (7.1) | 16 (4.4) | 74 (5.6) | < .001 | 189 (6.7) | n/a |
| Preterm labour (<37 weeks) | 36 (7.1) | 29 (4.8) | 11 (3.0) | 73 (5.5) | .092 | 149 (5.3) | n/a |
| Membranes ruptured in absence of labour | 41 (8.0) | 33 (5.4) | 30 (8.3) | 70 (5.3) | .069 | 174 (6.2) | n/a |

*Note*: n/a = not available.

[a] Frequencies do not sum to the total due to a small amount of missing data (n = 11)

[1] Based on Queensland Perinatal Statistics annual report for 2011 (n = 61125) [29]

[2] Figures for the Queensland population sample sourced from Australia's Mothers & Babies report (2011, n = 61112) [30]

likely to have an unassisted vaginal birth (70.2%) compared to women who received Standard Public Care (56.5%, OR 1.87, 99% CI 1.19–2.93) and less likely to have a scheduled caesarean birth (4.1% vs. 13.5%, OR 0.36, 99% CI 0.15–0.87) (Table 4). Women in the Public Midwifery Continuity Model were also less likely to experience breastfeeding problems (43.2% vs. 55.8%, OR 0.56, 99% CI 0.38–0.82), depression after birth (21.7% vs. 32.0%, OR 0.62, 99% CI 0.40–0.96) and spent fewer nights in hospital following birth (2.05 nights vs. 2.67 nights) than women in Standard Public Care. Infants born to women in Public Midwifery Continuity Care were less likely to be preterm (<37 weeks; 3.0% vs. 9.4%, OR 0.39, 99% CI 0.16–0.98) or be admitted to the NICU (12.2% vs. 24.9%, OR 0.49, 99% CI 0.29–0.81) and had higher odds of breastfeeding at 13 weeks of age (82.0% vs. 65.3%, 2.06, 99% CI 1.31–3.24). Women in the Public Midwifery Continuity model were also less likely to have an episiotomy (6.6% vs. 11%, OR 0.51, 0.26–0.99) or give birth to a baby with low birth weight (<2500 grams, 1.9% vs. 5.7%, OR 0.32, 99% CI 0.10–0.96) after initial adjustments for sociodemographic and reproductive characteristics. However, these associations were no longer significant after further adjustments for clinical covariates (episiotomy: OR 0.51, 99% CI 0.24–1.09; infant low birth weight: OR 0.43, 99% CI 0.14–1.38) (Table 4). After further adjustments for parity and pregnancy complications, women in the Public Midwifery Continuity model had lower odds of having an assisted vaginal birth (8.3% vs. 12.0%, OR 0.50, 99% CI 0.26–0.94) and experiencing anxiety after birth (42.3% vs. 50.4%, OR 0.68, 99% CI 0.47–0.99). After further adjustment for mode of birth,

**Table 4. Adjusted odds ratios for obstetric interventions and maternal and infant health outcomes by maternity model of care, adjusting for (i) sociodemographic characteristics, reproductive history, and (ii) with additional adjustment for relevant clinical covariates.**

| | GP Shared Care[1] | | Public Midwifery Continuity Care[1] | | Private Obstetric Care[1] | |
|---|---|---|---|---|---|---|
| | aOR(i) [99% CI] | aOR(ii) [99% CI] | aOR(i) [99% CI] | aOR(ii) [99% CI] | aOR(i) [99% CI] | aOR(ii) [99% CI] |
| **Maternal Outcomes** | | | | | | |
| Mode of birth | | | | | | |
| Unassisted vaginal birth | 1.13 [0.80–1.59] | 1.24[a] [0.85–1.83] | 1.57 [1.05–2.35]* | 1.87[a] [1.19–2.93]** | 0.56 [0.41–0.77]** | 0.63[a] [0.44–0.90]** |
| Assisted vaginal birth | 0.98 [0.61–1.60] | 0.90[a] [0.55–1.49] | 0.56 [0.30–1.04] | 0.50[a] [0.26–0.94]* | 1.25 [0.81–1.93] | 1.06[a] [0.68–1.66] |
| Scheduled caesarean birth | 1.24 [0.70–2.18] | 1.27[a] [0.71–2.27] | 0.36 [0.15–0.86]* | 0.36[a] [0.15–0.87]* | 3.87 [2.35–6.40]** | 3.73[a] [2.24–6.21]** |
| Unscheduled caesarean birth | 0.72 [0.47–1.12] | 0.68[a] [0.43–1.08] | 0.98 [0.61–1.58] | 0.95[a] [0.57–1.57] | 0.72 [0.48–1.07] | 0.59[a] [0.39–0.90]* |
| Induction of labour | 0.86 [0.59–1.23] | 0.92[b] [0.63–1.34] | 0.76 [0.49–1.15] | 0.78[b] [0.50–1.20] | 1.45 [1.04–2.03]* | 2.10[b] [1.47–2.99]** |
| Epidural/spinal block during labour | 0.76 [0.52–1.10] | 0.73[b] [0.49–1.08] | 0.80 [0.52–1.22] | 0.70[b] [0.44–1.09] | 1.09 [0.78–1.52] | 1.32[b] [0.92–1.90] |
| Continuous fetal monitoring during labour | 1.03 [0.74–1.42] | 1.05[b] [0.74–1.49] | 0.76 [0.52–1.10] | 0.69[b] [0.46–1.02] | 0.95 [0.71–1.29] | 1.35[b] [0.97–1.88] |
| Vaginal examinations during labour | 0.88 [0.64–1.20] | 0.88[b] [0.67–1.18] | 0.80 [0.55–1.15] | 0.69[b] [0.49–0.96]* | 0.66 [0.49–0.88]** | 0.88[b] [0.67–1.15] |
| Perineal status | | | | | | |
| Perineum intact | 0.94 [0.68–1.31] | 1.20[c] [0.77–1.87] | 1.17 [0.80–1.69] | 1.60[c] [0.98–2.60] | 1.21 [0.89–1.64] | 0.97[c] [0.63–1.51] |
| Perineal trauma with no sutures | 0.80 [0.44–1.46] | 0.75[c] [0.40–1.40] | 1.63 [0.89–2.97] | 1.51[c] [0.80–2.83] | 0.37 [0.20–0.70]** | 0.47[c] [0.24–0.90]* |
| Sutured perineal trauma | 1.14 [0.82–1.59] | 0.97[c] [0.64–1.47] | 0.70 [0.48–1.03] | 0.53[c] [0.33–0.84]** | 1.02 [0.75–1.39] | 1.36[c] [0.90–2.05] |
| Type of perineal trauma | | | | | | |
| Episiotomy | 1.09 [0.67–1.80] | 1.04[c] [0.58–1.85] | 0.51 [0.26–0.99]* | 0.51[c] [0.24–1.09] | 1.51 [0.96–2.36] | 1.69[c] [0.99–2.89] |
| Perineal tear | 1.03 [0.74–1.44] | 0.90[c] [0.61–1.32] | 0.97 [0.67–1.42] | 0.78[c] [0.50–1.21] | 0.72 [0.53–0.98]* | 0.88[c] [0.60–1.28] |
| Perineal tear following episiotomy | 1.08 [0.51–2.32] | 1.12[c] [0.50–2.52] | 0.45 [0.15–1.35] | 0.48[c] [0.16–1.51] | 1.45 [0.73–2.91] | 1.54[c] [0.74–3.23] |
| Maternal length of hospital stay (nights) | 0.88 [0.69–1.10] | 0.89[d] [0.73–1.10] | 0.57 [0.44–0.75]** | 0.61[d] [0.48–0.77]** | 4.61 [3.72–5.70]* | 3.94[d] [3.25–4.77]* |
| Experienced breastfeeding problems | 0.93 [0.67–1.28] | 0.92[e] [0.66–1.27] | 0.56 [0.39–0.81]** | 0.56[e] [0.38–0.82]** | 0.98 [0.73–1.32] | 0.93[e] [0.69–1.26] |
| Experienced depression after birth | 0.77 [0.55–1.09] | 0.73[a] [0.51–1.04] | 0.61 [0.40–0.93]* | 0.62[a] [0.40–0.96]* | 0.84 [0.62–1.15] | 0.84[a] [0.61–1.16] |
| Experienced anxiety after birth | 0.85 [0.62–1.16] | 0.80[a] [0.58–1.11] | 0.70 [0.49–1.01] | 0.68[a] [0.47–0.99]* | 0.98 [0.74–1.31] | 0.93[a] [0.69–1.25] |
| Diagnosed depression after birth | 0.63 [0.36–1.11] | 0.55[a] [0.30–1.02] | 0.73 [0.38–1.39] | 0.81[a] [0.41–1.62] | 0.60 [0.36–1.01] | 0.65[a] [0.37–1.13] |
| Diagnosed anxiety after birth | 0.62 [0.35–1.13] | 0.55[a] [0.29–1.03] | 0.63 [0.31–1.28] | 0.67[a] [0.32–1.42] | 0.67 [0.40–1.12] | 0.67[a] [0.38–1.17] |
| Maternal hospital re-admission | 0.98 [0.47–2.04] | 0.93[a] [0.45–1.95] | 0.62 [0.24–1.59] | 0.61[a] [0.24–1.57] | 0.96 [0.48–1.90] | 0.92[a] [0.46–1.84] |
| **Infant Outcomes** | | | | | | |
| Preterm birth (<37 weeks) | 0.51 [0.27–0.95]* | 0.58[b] [0.30–1.11] | 0.32 [0.13–0.79]* | 0.39[b] [0.16–0.98]* | 0.81 [0.48–1.35] | 0.95[b] [0.55–1.64] |
| Low infant birth weight (<2500g) | 0.84 [0.42–1.68] | 1.07[b] [0.50–2.29] | 0.32 [0.10–0.96]* | 0.43[b] [0.14–1.38] | 0.67 [0.35–1.31] | 0.83[b] [0.40–1.71] |
| Neonate admission to NICU | 0.63 [0.43–0.92]* | 0.68[b] [0.45–1.01] | 0.44 [0.27–7.30]** | 0.49[b] [0.29–0.81]** | 0.48 [0.33–0.69]** | 0.49[b] [0.33–0.72]** |
| Neonate's length of stay in NICU† | | | | | | |
| < 48 hours | 2.01 [0.97–4.16] | 1.93[b] [0.90–4.13] | 1.22 [0.46–3.21] | 1.13[b] [0.42–3.10] | 1.29 [0.66–2.53] | 1.29[b] [0.63–2.61] |
| 48 hours to 7 days | 0.45 [0.21–0.96]* | 0.41[b] [0.19–0.90]* | 1.11 [0.43–2.88] | 1.10[b] [0.41–2.92] | 0.48 [0.24–0.95]* | 0.45[b] [0.22–0.92]* |
| > 7 days | 1.09 [0.45–2.59] | 1.27[b] [0.49–3.30] | 0.59 [0.16–2.17] | 0.60[b] [0.15–2.40] | 1.77 [0.83–3.81] | 2.05[b] [0.87–4.80] |
| Infant hospital re-admission | 0.91 [0.52–1.61] | 0.93[a] [0.52–1.66] | 1.13 [0.60–2.11] | 1.20[a] [0.64–2.27] | 0.74 [0.43–1.27] | 0.81[a] [0.47–1.40] |
| Breastfeeding at 13 weeks | 1.08 [0.76–1.52] | 1.05[e] [0.74–1.50] | 2.14 [1.37–3.35]** | 2.06[e] [1.31–3.24]** | 1.31 [0.95–1.83] | 1.33[e] [0.95–1.86] |

*Note.* Adjustments for sociodemographic characteristics and reproductive history include: maternal age, BMI, area of residence, education, language spoken at home, country of birth and previous caesarean; aOR = adjusted odds ratio; NICU = neonatal intensive care unit.

[1] vs. Standard Public Care

[a] adjusted for sociodemographic characteristics, reproductive history, parity and complications during index pregnancy (depression, gestational diabetes, amount of amniotic fluid was a concern, baby was too small).

[b] adjusted for sociodemographic characteristics, reproductive history, parity, complications during index pregnancy (depression, gestational diabetes, amount of amniotic fluid was a concern, baby was too small) and mode of birth (vaginal birth, scheduled caesarean birth or unscheduled caesarean birth).

[c] adjusted for sociodemographic characteristics, reproductive history, parity, complications during index pregnancy (depression, gestational diabetes, amount of amniotic fluid was a concern, baby was too small) and mode of birth (unassisted vaginal birth, assisted vaginal birth or caesarean birth).

[d] adjusted for sociodemographic characteristics, reproductive history, parity, complications during index pregnancy (depression, gestational diabetes, amount of amniotic fluid was a concern, baby was too small) and mode of birth (vaginal birth or caesarean birth).

[e] adjusted for sociodemographic characteristics, reproductive history, parity, complications during index pregnancy (depression, gestational diabetes, amount of amniotic fluid was a concern, baby was too small), mode of birth (unassisted vaginal birth, assisted vaginal birth or caesarean birth) and neonate admission to NICU.

* $p < .01$

** $p < .001$

† Of the neonates admitted to the NICU ($n = 456$).

women in the Public Midwifery Continuity Model were also significantly less likely to have perineal trauma that required sutures (35.5% vs. 38.4%, OR 0.53, 99% CI 0.33–0.84) and had less vaginal examinations during labour.

Women who received Private Obstetric Care had 3.73 [99% CI 2.24–6.21] times higher odds of having a scheduled caesarean birth (31.4%) than those who received Standard Public Care (13.5%) and were significantly less likely to have an unassisted vaginal birth (40.7% vs. 56.5%, OR 0.63, 99% CI 0.44–0.90) (Table 4). They had 2.10 [99% CI 1.47–2.99] times higher odds of being induced (30.9% vs. 28.2%) and had a longer hospital stay after birth (4.31 nights vs. 2.67 nights). Infants born to women who received care in the Private Obstetric model had lower odds of an admission to the NICU (13.6% vs. 24.9%, OR 0.48, 99% CI 0.33–0.69). These associations were significant across all adjusted models. Women in the Private Obstetric model were also less likely to have a perineal tear (32.7% vs. 40.0%, OR 0.72, 99% CI 0.52–0.98) and had fewer vaginal examinations during labour after initial adjustments for sociodemographic and reproductive characteristics. However, these associations were no longer significant after further adjustments for clinical covariates. In the final adjusted model, women in Private Obstetric Care were also significantly less likely to have an unscheduled caesarean birth (12.8% vs. 18.0%, OR 0.59, 99% CI 0.39–0.90) (Table 4).

Infants born to women who received GP Shared Care had significantly lower odds of admission to NICU (17.4% vs. 24.9%, OR 0.51, 99% CI 0.27–0.95) and preterm birth (4.9% vs. 9.4%, OR 0.63, 99% CI 0.43–0.92) after adjustments for sociodemographic and reproductive characteristics. However, these associations were no longer significant after further adjustments for clinical covariates (NICU admission: OR 0.68, 99% CI 0.45–1.01; preterm birth: OR 0.58, 99% CI 0.30–1.11) (Table 4). The odds of all other obstetric intervention and maternal and infant health outcomes did not significantly differ between GP Shared Care and Standard Public Care. A graphical summary of findings for all maternal and infant outcomes where a significant difference (p<0.001) was found between Standard Public Care and at least one MMC are shown in Fig 1.

### Information provision and decision-making

Information provision for procedures during pregnancy did not differ between any MMC when compared to Standard Public Care, nor did experiencing procedures during pregnancy

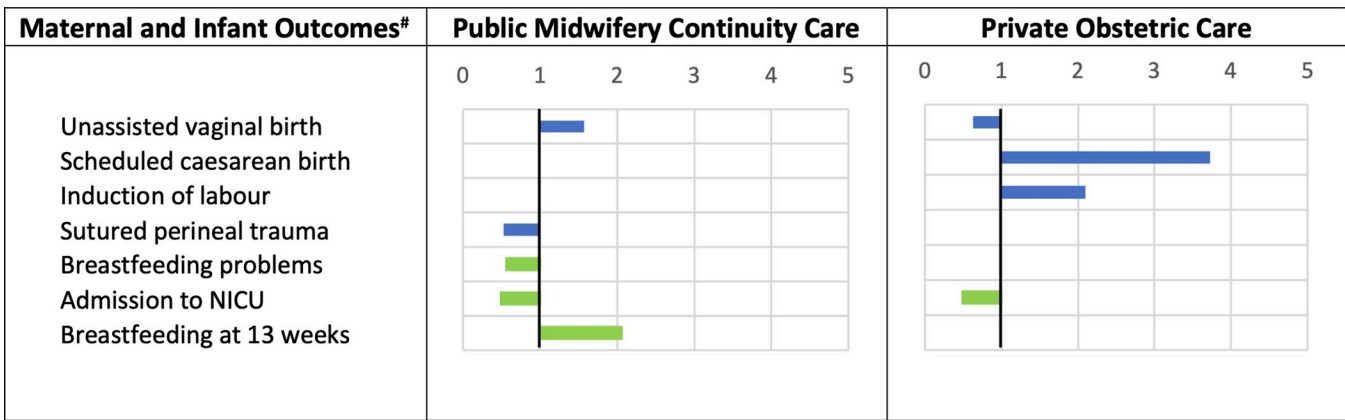

*Final adjustment for confounders as per Tables 4, 5, 6 and 7

#p< 0.001; no significant differences between GP Shared Care and Standard Public Care

**Fig 1. Adjusted odds\* of maternal and infant outcomes significantly different to Standard Public Care#.** Green indicates improved outcomes, red indicates poorer outcomes and blue indicates outcomes with differential value dependent on consumer needs and preferences.

**Table 5. Adjusted odds ratios for information provision and decision-making by maternity model of care, adjusting for (i) sociodemographic characteristics, reproductive history, and (ii) with additional adjustment for relevant clinical covariates.**

| | GP Shared Care[1] | | Public Midwifery Continuity Care[1] | | Private Obstetric Care[1] | |
|---|---|---|---|---|---|---|
| | aOR[(i)] [99% CI] | aOR[(ii)] [99% CI] | aOR[(i)] [99% CI] | aOR[(ii)] [99% CI] | aOR[(i)] [99% CI] | aOR[(ii)] [99% CI] |
| **Pros and cons discussed for having/not having** | | | | | | |
| Ultrasound scans | 1.26 [0.92–1.73] | 1.25[a] [0.91–1.72] | 1.16 [0.81–1.67] | 1.14[a] [0.80–1.65] | 1.12 [0.84–1.49] | 1.11[a] [0.83–1.49] |
| Blood tests during pregnancy | 1.26 [0.91–1.75] | 1.25[a] [0.90–1.73] | 1.29 [0.87–1.89] | 1.26[a] [0.86–1.85] | 1.35 [1.00–1.82] | 1.34[a] [0.99–1.81] |
| Caesarean birth | 1.23 [0.88–1.73] | 1.21[a] [0.85–1.73] | 1.21 [0.82–1.77] | 1.19[a] [0.80–1.79] | 1.65 [1.20–2.27]** | 1.50[a] [1.07–2.10]* |
| Induction of labour | 0.94 [0.67–1.33] | 0.93[a] [0.65–1.31] | 1.77 [1.14–2.74]** | 1.74[a] [1.12–2.71]* | 1.01 [0.74–1.39] | 0.98[a] [0.71–1.35] |
| Fetal monitoring during labour | 1.05 [0.74–1.50] | 1.05[a] [0.74–1.50] | 1.68 [1.08–2.60]* | 1.68[a][1.08–2.61]* | 0.80 [0.58–1.09] | 0.77[a] [0.56–1.06] |
| Vaginal examinations | 1.22 [0.88–1.70] | 1.20[a] [0.86–1.68] | 2.08 [1.39–3.13]** | 2.05[a][1.36–3.09]** | 0.87 [0.65–1.18] | 0.84[a] [0.62–1.14] |
| Epidural | 1.03 [0.68–1.55] | 0.98[a] [0.64–1.49] | 1.17 [0.73–1.90] | 1.14[a] [0.69–1.89] | 1.30 [0.89–1.91] | 1.13 [0.76–1.69] |
| Episiotomy | 1.14 [0.82–1.57] | 1.09[a] [0.79–1.51] | 1.55 [0.17–2.25]* | 1.48[a] [1.01–2.16]* | 0.92 [0.69–1.24] | 0.83[a] [0.62–1.13] |
| Syntocinon to birth placenta | 1.22 [0.86–1.72] | 1.19[a] [0.84–1.69] | 2.32 [1.49–3.64]** | 2.24[a] [1.43–3.51]** | 0.52 [0.38–0.72]** | 0.50[a] [0.36–0.68]** |
| **Procedures experienced without consent** | | | | | | |
| Ultrasound scans | 0.58 [0.29–1.67] | 0.60[a][0.30–1.21] | 1.10 [0.55–2.21] | 1.17[a][0.58–2.35] | 1.01 [0.58–1.78] | 1.05[a] 0.59–1.85] |
| Blood tests during pregnancy | 0.81 [0.47–1.40] | 0.85[a] [0.49–1.48] | 0.78 [0.41–1.49] | 0.85[a] [0.45–1.63] | 0.79 [0.49–1.69] | 0.84[a] [0.51–1.38] |
| Caesarean birth | 0.78 [0.49–1.24] | 0.78[b] [0.49–1.25] | 0.48 [0.26–0.87]* | 0.48[b] [0.26–0.88]* | 0.19 [0.11–0.33]** | 0.21[b] [0.12–0.36]** |
| Induction of labour | 0.79 [0.44–1.42] | 0.81[b] [0.45–1.48] | 0.72 [0.35–1.46] | 0.80[b] [0.39–1.64] | 0.55 [0.32–0.96]* | 0.56[b] [0.31–0.99]* |
| Missing data | 0.88 [0.42–1.86] | 0.93[b] [0.42–2.05] | 0.43 [0.14–1.34] | 0.59[b] [0.18–1.48] | 1.10 [0.58–2.10] | 0.74 [0.37–1.48] |
| Fetal monitoring during labour | 0.99 [0.68–1.45] | 1.01[b] [0.68–1.49] | 0.52 [0.32–0.85]** | 0.50[b] [0.31–0.82]** | 0.74 [0.52–1.06] | 0.90[b] [0.62–1.30] |
| Vaginal examinations | 0.90 [0.52–1.55] | 0.89[b] [0.51–1.55] | 0.42 [0.19–0.91]* | 0.40[b] [0.18–0.87]* | 0.87 [0.53–1.44] | 1.01[b] [0.61–1.68] |
| Epidural | 0.82 [0.44–1.55] | 0.87[c] [0.46–1.65] | 0.56 [0.24–1.29] | 0.66[c] [0.28–1.54] | 0.67 [0.37–1.22] | 0.68[c] [0.37–1.25] |
| Episiotomy | 1.06 [0.70–1.62] | 1.05[c] [0.67–1.64] | 0.57 [0.34–0.97]* | 0.57[c] [0.33–1.00] | 0.96 [0.65–1.43] | 1.10[c] [0.72–1.68] |
| Syntocinon to birth placenta | 1.25 [0.73–2.14] | 1.22[d] [0.71–2.11] | 1.12 [0.61–2.06] | 1.09[d] [0.59–2.04] | 1.91 [1.18–3.09]** | 2.34[d] [1.42–3.85]** |

*Note*: Adjustments for sociodemographic characteristics and reproductive history include: maternal age, BMI, area of residence, education, language spoken at home, country of birth and previous caesarean; aOR = adjusted odds ratio.

[1] vs. Standard Public Care

[a] adjusted for sociodemographic characteristics, reproductive history, parity and complications during index pregnancy (depression, gestational diabetes, amount of amniotic fluid was a concern, baby was too small).

[b] adjusted for sociodemographic characteristics, reproductive history, parity, complications during index pregnancy (depression, gestational diabetes, amount of amniotic fluid was a concern, baby was too small) and mode of birth (vaginal birth, scheduled caesarean birth or unscheduled caesarean birth).

[c] adjusted for sociodemographic characteristics, reproductive history, parity, complications during index pregnancy (depression, gestational diabetes, amount of amniotic fluid was a concern, baby was too small) and mode of birth (unassisted vaginal birth, assisted vaginal birth or caesarean birth).

[d] adjusted for sociodemographic characteristics, reproductive history, parity, complications during index pregnancy (depression, gestational diabetes, amount of amniotic fluid was a concern, baby was too small) and mode of birth (vaginal birth or caesarean birth).

* $p < .01$

** $p < .001$

without consent. However, women who received Public Midwifery Continuity Care and Private Obstetric Care were both significantly more likely to have their care providers discuss with them the pros and cons of various specific obstetric procedures during labour/birth than women who received Standard Public Care. Women in the Public Midwifery Continuity model were significantly more likely to have their care providers discuss with them the pros and cons of having and not having an induction of labour (81.5% vs. 69.4%, OR 1.74, 99% CI 1.12–2.71), fetal monitoring (81.5% vs. 72.7%, OR 1.68, 99% CI 1.08–2.61), vaginal examinations (76.2% vs. 60.8%, OR 2.05, 99% CI 1.36–3.09), an episiotomy (60.5% vs. 47.5%, OR 1.48, 99% CI 1.01–2.16) and administering Syntocinon to birth their placenta (82.9% vs. 64.9%, OR 2.24, 99% CI 1.43–3.51) (Table 5). Women in the Private Obstetric model were more likely to

have their care providers discuss the pros and cons of having and not having a caesarean birth (75.2% vs. 643.7%, OR 1.50, 99% CI 1.07–2.10), and less likely to have their care providers discuss with them the pros and cons of having or not having Syntocinon to birth their placenta (49.0% vs. 64.9%, OR 0.50, 99% CI 0.36–0.68) (Table 5). These associations were significant across all adjusted models.

Women in Public Midwifery Continuity Care were significantly less likely than those in Standard Public Care to have had several procedures without their consent, including caesarean birth (8.6% vs. 14.7%, OR 0.48, 99% CI 0.26–0.88), vaginal examinations during labour (4.4% vs. 9.4%, OR 0.40, 99% CI 0.18–0.87), and fetal monitoring during labour (14.1% vs. 22.2%, OR 0.50, 99% CI 0.31–0.82) (Table 1). Women in Private Obstetric Care were significantly less likely than those in Standard Public Care to have had a caesarean birth without their consent (3.2% vs. 14.7%, OR 0.21, 99% CI 0.12–0.36) and induction of labour without consent (5.4% vs. 8.6%, OR 0.56, 99% CI 0.31–0.99), but had 2.34 [99% CI 1.42–3.85] times higher odds of having had Syntocinon administered to birth their placenta without their consent (15.4% vs. 8.6%) (Table 1). These associations were significant across all adjusted models. Women in the Public Midwifery Continuity Model were also less likely to experience an episiotomy without their consent (11.9% vs. 16.5%, OR 0.57, 99% CI 0.34–0.97) after adjusting for sociodemographic and reproductive characteristics, however, this was not significant after further adjustments for clinical covariates (OR 0.57, 99% CI 0.33–1.00) (Table 1). There were no significant differences in information provision or consent to obstetric interventions between GP Shared Care and Standard Public Care.

**Maternal care experiences during pregnancy, labour/birth, and postpartum.** After all adjustments, women in GP Shared Care and Private Obstetric Care had a significantly earlier first pregnancy check-up (8.08 weeks and 7.74 weeks respectively) and a significantly later booking appointment at their planned place of birth (17.74 weeks and 17.48 weeks respectively) than women in Standard Public Care (9.07 weeks at first pregnancy check-up and 16.26 weeks at booking appointment) (Table 6). Women in GP Shared Care (11.66 check-ups) and Private Obstetric Care (11.95 check-ups) had significantly more pregnancy check-ups in comparison to Standard Public Care (10.94 check-ups). Women in Public Midwifery Continuity Care were initially observed to have fewer pregnancy check-ups (9.99 vs.10.94), however, this association was no longer significant after additional adjustments for pregnancy complications and parity.

Women who received Private Obstetric Care had 2.07 [99% CI 1.38–3.1] times higher odds of being satisfied with the timing of their booking appointment (88.9% vs. 77.6%) and 1.99 [99% CI 1.31–3.03] times higher odds of being satisfied with the number of pregnancy check-ups (90.7% vs. 81.8%) and 2.72 [99% CI 2.01–3.68] times higher odds of being able to choose their mode of birth (75.0% vs. 49.8%). Women had significantly lower odds of being able to choose their mode of birth if they received Public Midwifery Continuity Care (39.5% vs. 49.8%, OR 0.63, 99% CI 0.44–0.92). Women had 2.18 [99% CI 1.17–4.07] times higher odds of being able to choose the gender of their care provider if they received Public Midwifery Continuity Care (13.3% vs. 6.5%) and 6.94 [99% CI 4.11–11.72] times higher odds if they received Private Obstetric Care (29.6% vs. 6.5%). Women had 4.70 [99% CI 3.16–6.98] times higher odds of having one person coordinating their pregnancy care in GP shared Care (86.4%), 4.98 [99% CI 3.05–8.14] times higher odds in Public Midwifery Continuity Care (88.1%) and 36.49 [99% CI 20.48–65.04] times higher odds in Private Obstetric Care (98.0%) compared to women who received Standard Public Care (58.2%). Women in the Public Midwifery Continuity and Private Obstetric Care models were more likely to be given the after-hours contact details of a named care provider during their pregnancy (67.4%, OR 6.36, 99% CI 2.16–18.68 and 74.3%, OR 5.29, 99% CI 2.59–10.82 respectively vs. 25.9% in Standard Public Care). These associations were significant across all adjusted models.

**Table 6. Adjusted odds ratios for maternal experiences during pregnancy, labour/birth, and postpartum care by maternity model of care, adjusting for (i) sociodemographic characteristics, reproductive history, and (ii) with additional adjustment for relevant clinical covariates.**

| | GP Shared Care[1] | | Public Midwifery Continuity Care[1] | | Private Obstetric Care[1] | |
|---|---|---|---|---|---|---|
| | aOR[(i)] [99% CI] | aOR[(ii)] [99% CI] | aOR[(i)] [99% CI] | aOR[(ii)] [99% CI] | aOR[(i)] [99% CI] | aOR[(ii)] [99% CI] |
| **Pregnancy** | | | | | | |
| Able to choose gender of care provider | 1.06 [0.56–1.98] | 1.05[a] [0.56–1.97] | 2.14 [1.15–3.99]* | 2.18[a] [1.17–4.07]* | 6.81 [4.05–11.46]** | 6.94[a] [4.11–11.72]** |
| Able to choose mode of birth | 1.01 [0.74–1.38] | 0.99[a] [0.72–1.36] | 0.65 [0.45–0.94]* | 0.63[a] [0.44–0.92]* | 2.84 [2.10–3.84]** | 2.72[a] [2.01–3.68]** |
| Weeks gestation at first pregnancy check-up$ | 0.42 [0.23–0.76]** | 0.43[a][0.24–0.77]** | 1.11 [0.56–2.17] | 1.15[a] [0.59–2.26] | 0.34 [0.20–0.58]** | 0.37[a] [0.22–0.64]** |
| Number of pregnancy check-ups | 2.28 [1.12–4.62]* | 2.58[a] [1.28–5.18]** | 0.44 [0.19–0.99]* | 0.54[a] [0.24–1.21] | 3.91 [2.04–7.47]** | 4.37[a] [2.3–8.32]** |
| Satisfied with number of pregnancy check-ups | 1.36 [0.88–2.09] | 1.34[a] [0.87–2.07] | 1.66 [0.99–2.81] | 1.61[a] [0.95–2.73] | 2.10 [1.38–3.17]** | 1.99[a] [1.31–3.03]** |
| Weeks gestation at booking appointment† | 6.11 [1.95–19.12]** | 5.91[a] [1.89–18.53]** | 0.78 [0.21–2.94] | 0.75[a] [1.20–2.82] | 4.90 [1.68–14.29]** | 5.01[a] [1.71–14.73]** |
| Satisfied with timing of booking appointment† | 1.05 [0.72–1.55] | 1.05[a] [0.71–1.55] | 1.15 [0.73–1.81] | 1.13[a] [0.72–1.79] | 2.13 [1.43–3.17]** | 2.07[a] [1.38–3.10]** |
| One person coordinating pregnancy care | 4.75 [3.20–7.05]** | 4.70[a][3.16–6.98]** | 5.08 [3.12–8.29]** | 4.98[a][3.05–8.14]** | 36.80 [20.69–65.45]** | 36.49[a] [20.48–65.04]** |
| Given after hours contact details of a care provider | | | | | | |
| A named care provider | 1.19 [0.61–2.36] | 1.21[a] [0.61–2.39] | 6.51 [2.23–19.05]** | 6.36[a] [2.16–18.68]** | 5.50 [2.71–11.19]** | 5.29[a] [2.59–10.82]** |
| A hospital, clinic, or health service | 0.67 [0.35–1.29] | 0.68[a] [0.35–1.31] | 1.17 [0.40–3.43] | 1.14[a] [0.39–3.35] | 0.69 [0.34–1.38] | 0.67[a] [0.33–1.43] |
| **Labour/Birth** | | | | | | |
| A known care provider during labour/birth | 0.59 [0.43–0.82]** | 0.60[a] [0.44–0.83]** | 4.26 [2.79–6.51]** | 4.35[a] [2.84–6.67]** | 8.41 [5.91–11.98]** | 8.43[a] [5.90–12.05]** |
| Continuity of care throughout labour/birth | 1.03 [0.75–1.42] | 1.06[a] [0.77–1.46] | 2.37 [1.59–3.53]** | 2.47[a] [1.65–3.71]** | 4.37 [3.15–6.06]** | 4.71[a] [3.37–6.57]** |
| Mobility during labour | 1.20 [0.86–1.68] | 1.21[b] [0.85–1.73] | 2.10 [1.43–3.07]** | 2.06[b][1.37–3.08]** | 0.75 [0.54–1.03] | 0.93[b] [0.66–1.31] |
| Support people made to feel welcome | | | | | | |
| During labour | 1.06 [0.64–1.76] | 1.17[b] [0.64–2.14] | 2.02 [0.97–4.19] | 1.44[b][0.62–3.34] | 0.58 [0.38–0.91]* | 1.13[b] [0.66–1.93] |
| During birth | 0.67 [0.29–1.55] | 0.58[b] [0.24–1.40] | 1.05 [0.36–3.07] | 0.97[b] [0.32–2.96] | 1.70 [0.67–4.33] | 1.39[b] [0.53–3.63] |
| Skin-to-skin contact first time holding baby | 1.08 [0.74–1.59] | 1.00[c] [0.64–1.55] | 1.84 [1.11–3.06]* | 1.72[c] [0.87–3.04] | 0.66 [0.47–0.94]* | 0.81[c] [0.55–1.20] |
| Perceived all medical procedures necessary‡ | 1.47 [0.93–2.33] | 1.47[a][0.93–2.34] | 1.00 [0.59–1.68] | 1.00[a] [0.59–1.69] | 2.08 [1.35–3.21]** | 2.08[a] [1.35–3.22]** |
| **Postpartum Care** | | | | | | |
| Support people were made to feel welcome | | | | | | |
| After birth | 0.95 [0.49–1.87] | 1.00[a] [0.51–1.96] | 2.71 [0.94–7.81] | 2.83[a] [0.98–8.20] | 3.74 [1.66–8.42]** | 4.10[a] [1.81–9.27]** |
| Overnight ⌐ | 1.07 [0.72–1.59] | 1.08[a] [0.73–1.61] | 1.85 [1.18–2.91]** | 1.88[a] [1.19–2.96]** | 27.46 [18.01–41.87]** | 27.62[a] [18.05–42.25]** |
| Satisfied with length of hospital stay | 1.25 [0.87–1.78] | 1.25[a] [0.87–1.79] | 1.30 [0.86–1.97] | 1.25[a] [0.83–1.91] | 2.40 [1.70–3.40]** | 2.41[a] [1.70–3.43]** |
| Given after hours contact details of a care provider | | | | | | |
| A named care provider | 2.37 [1.15–4.89]* | 2.47[a] [1.20–5.10]** | 4.96 [1.95–12.61]** | 5.03[a] [1.97–12.86]** | 2.37 [1.26–4.44]** | 2.48[a] [1.31–4.67]** |
| A hospital, clinic, or health service | 1.48 [0.74–2.99] | 1.54[a] [0.76–3.11] | 1.05 [0.41–2.70] | 1.07[a] [0.41–2.74] | 0.74 [0.40–1.36] | 0.76[a][0.41–1.41] |
| Visited at home or telephoned after arriving home | 1.14 [0.62–2.11] | 1.14[a] [0.62–2.11] | 4.98 [1.57–15.74]** | 4.89[a] [1.54–15.50]** | 0.02 [0.01–0.04]** | 0.02[a] [0.01–0.04]** |
| Missing data | 1.56 [0.54–4.48] | 1.59[a] [0.55–4.59] | 6.70 [1.48–30.26]* | 6.68[a] [1.47–30.37]* | 0.08 [0.03–0.22]** | 0.08 [0.03–0.22]** |

*(Continued)*

**Table 6.** (*Continued*)

| | GP Shared Care[1] | | Public Midwifery Continuity Care[1] | | Private Obstetric Care[1] | |
|---|---|---|---|---|---|---|
| | aOR[(i)] [99% CI] | aOR[(ii)] [99% CI] | aOR[(i)] [99% CI] | aOR[(ii)] [99% CI] | aOR[(i)] [99% CI] | aOR[(ii)] [99% CI] |
| Confident to care for baby at home after birth | 1.42 [0.86–2.36] | 1.62[a] [0.95–2.78] | 1.32 [0.76–2.32] | 1.40[a] [0.77–2.54] | 1.00 [0.65–1.55] | 1.21[a] [0.76–1.93] |

*Note*: Adjustments for sociodemographic characteristics and reproductive history include: maternal age, BMI, area of residence, education, language spoken at home, country of birth and previous caesarean; aOR = adjusted odds ratio

[1] vs. Standard Public Care

[a] adjusted for sociodemographic characteristics, reproductive history, parity and complications during index pregnancy (depression, gestational diabetes, amount of amniotic fluid was a concern, baby was too small).

[b] adjusted for sociodemographic characteristics, reproductive history, parity, complications during index pregnancy (depression, gestational diabetes, amount of amniotic fluid was a concern, baby was too small) and mode of birth (vaginal birth, scheduled caesarean birth or unscheduled caesarean birth).

[c] adjusted for sociodemographic characteristics, reproductive history, parity, complications during index pregnancy (depression, gestational diabetes, amount of amniotic fluid was a concern, baby was too small) and mode of birth (unassisted vaginal birth, assisted vaginal birth or caesarean birth).

\* $p < .01$

\*\* $p < .001$

† Of the women who had a booking appointment (*n* = 2469)

‡ Of the women who had medical procedures (*n* = 2298)

§ Of the women who had pregnancy check-ups (*n* = 2796)

ɕ Of the women who stayed overnight (*n* = 2306)

The odds of having a known care provider during labour and birth were 4.35 [2.84–6.67] times higher in Public Midwifery Continuity Care (80.4%) and 8.43 [99% CI 5.90–12.05] times higher in Private Obstetric Care (89.6%), but lower for GP Shared Care (39.4%, OR 0.60, 99% CI 0.44–0.83), compared to Standard Public Care (52.9%) (Table 6). Continuity of care during labour and birth was also significantly more likely in Public Midwifery Continuity (76.0% vs. 57.1%, OR 2.47, 99% CI 1.65–3.71) and Private Obstetric (85.4% vs. 57.1%, OR 4.71, 99% CI 3.37–6.57) models. Women who received Public Midwifery Continuity Care also had twice the odds of freedom of mobility during labour than women in Standard Public Care (55.2% vs. 34.7%, OR 2.06, 99% CI 1.37–3.08). Women in Private Obstetric Care were more likely to perceive all received medical procedures as necessary (90.2% vs. 78.8%, OR 2.08, 99% CI 1.35–3.21). These associations were significant across all adjusted models. Women in Public Midwifery Continuity Care were significantly more likely to have skin-to-skin contact the first time holding their baby (87.8% vs. 77.5%, OR 1.84, 99% CI 1.11–3.06), and women in the Private Obstetric model were significantly less likely (69.9% vs. 77.5%, OR 0.66, 99% CI 0.47–0.94) after initial adjustments for sociodemographic and reproductive history variables. However, these associations were no longer significant after further adjustments for clinical covariates (Public Midwifery Continuity Care: OR 1.72, 99% CI 0.87–3.04; Private Obstetric Care: OR 0.81, 0.55–1.20) (Table 6). The odds of having support people welcome during birth did not differ for any MMC compared to Standard Public Care.

After adjustments for sociodemographic characteristics, reproductive history, and relevant clinical covariates, women in Private Obstetric Care had 4.10 [99% CI 1.81–9.27] times higher odds of having their support people welcome after birth (98.5% vs. 94.3%) and 27.62 [99% CI 18.05–42.25] times higher odds of them feeling welcome overnight (91.2% vs. 30.2%) (Table 6). Women in GP Shared (41.9%, OR 2.47, 99% CI 1.20–5.10), Public Midwifery Continuity (68.0%, OR, 5.03, 99% CI 1.97–12.86) and Private Obstetric models (59.3%, OR 2.48, 99% CI 1.31–4.67) were more likely to be given the after-hours contact details of a named care provider at home after birth, than women who received Standard Public Care (29.8%). The

odds of having a nurse or midwife visit or telephone within the first seven days of arriving home after birth were substantially lower for women who received Private Obstetric Care (22.0% vs. 89.8%, OR 0.02, 99% CI 0.01–0.04) and 4.89 [99% CI 1.54–15.50] times higher for women who received Public Midwifery Continuity Care (94.5% vs. 89.8%). Women's confidence to care for their baby once going home did not differ between any MMC and Standard Public Care.

**Interpersonal and overall quality of care.** Women who received Public Midwifery Continuity Care were more likely to report better quality interpersonal and overall care across all care stages (ORs ranging between 1.51 [99% CI 1.02–2.23] and 3.19 [99% CI 2.00–5.08]), with the exception of involvement in decision-making during labour and postpartum care at home, which were not significantly different from Standard Public Care (Table 7). Women who received Private Obstetric Care were more likely to report better quality interpersonal and overall care across pregnancy, labour/birth and postpartum care in hospital (ORs ranging between 1.59 [99% CI 1.18–2.13] and 5.83 [99% CI 4.21–8.07]), with the exception of receiving conflicting information from different care providers where there was no significant difference (Table 7) Women in Private Obstetric Care also reported better interpersonal care across several care processes during postpartum care at home, including women feeling like care providers were on their side, teamwork, confidence in care provider skills and time to talk (ORs ranging between 1.39 [99% CI 1.02–1.87] and 1.46 [99% CI 1.07–1.98]) (Table 7). All other experiences of postpartum care at home in Private Obstetric Care were not significantly different from Standard Public Care. Women who received GP Shared Care were more likely to report having their care providers talk to them with kindness (73.9% vs. 65.7%, OR 1.51, 99% CI 1.06–2.13) and have care providers respect their decisions (73.7% vs. 65.5%, OR 1.45, 99% CI 1.03–2.05) during pregnancy, in comparison to Standard Public Care (Table 7). There were no other significant differences in interpersonal or overall quality of care between GP Shared and Standard Public Care. Significant results remained consistent across adjusted models.

A graphical summary of findings for all experiences where a significant difference (p<0.001) was found between Standard Public Care and at least one other MMC are shown in Fig 2.

## Discussion

There were differences observed between the broadly categorised MMCs compared here across most outcomes and experiential measures. This was largely due to differences between Public Midwifery Continuity Care or Private Obstetric Care, and Standard Public Care. Women are more commonly informed about only Standard Public Care and GP Shared Care during their first antenatal care visit [4], so these results highlight the importance of providing comparisons across all MMCs to ultimately improve informed decision-making about MMC for women. There were few differences between GP Shared Care and Standard Public Care. Differences between these two models were mostly seen in experiential measures within the antenatal care period. Women in GP Shared Care were more likely to have earlier first pregnancy check-ups and later booking appointments, one person coordinating pregnancy care and their care providers talk to them with kindness and understanding and respect their decisions. These results are not surprising, as continuity of antenatal care is the main point of difference between GP Shared Care and Standard Public Care [5]. Women in GP Shared Care were also less likely to have at least one known care provider during labour and birth. The primary care provider in GP Shared Care changes for the woman when she moves from antenatal to intrapartum care, compared to Standard Public Care where both antenatal and intrapartum care are provided by rostered doctors and midwifes who work in a public hospital [5]. This

**Table 7. Adjusted odds ratios for maternal experiences of interpersonal quality of care by maternity model of care, adjusting for (i) sociodemographic characteristics, reproductive history, and (ii) with additional adjustment for relevant clinical covariates.**

| | GP Shared Care[1] | | Public Midwifery Continuity Care[1] | | Private Obstetric Care[1] | |
|---|---|---|---|---|---|---|
| | aOR(i) [99% CI] | aOR(ii) [99% CI] | aOR(i) [99% CI] | aOR(ii) [99% CI] | aOR(i) [99% CI] | aOR(ii) [99% CI] |
| **Care providers communicated well with other care providers all of the time** | | | | | | |
| During pregnancy | 0.91 [0.65–1.27] | 0.90a [0.65–1.26] | 2.20 [1.52–3.18]** | 2.15a [1.48–3.12]** | 4.33 [3.19–5.89]** | 4.31a [3.16–5.87]** |
| During labour/birth | 0.93 [0.68–1.28] | 0.93a [0.67–1.28] | 2.05 [1.39–3.04]** | 2.03a [1.37–3.01]** | 2.50 [1.84–3.41]** | 2.50a [1.83–3.41]** |
| During postpartum care in hospital | 0.96 [0.69–1.32] | 0.96a [0.69–1.33] | 2.30 [1.59–3.33]** | 2.30a [1.58–3.33]** | 2.06 [1.53–2.77]** | 2.11a [1.56–2.85]** |
| During postpartum care after going home† | 0.80 [0.58–1.10] | 0.79a [0.57–1.10] | 2.05 [1.41–2.99]** | 2.00a [1.37–2.93]** | 1.30 [0.96–1.75] | 1.31a [0.97–1.77] |
| **Care providers worked well as a team all of the time** | | | | | | |
| During pregnancy | 1.02 [0.74–1.40] | 1.03a [0.73–1.39] | 3.23 [2.21–4.71]** | 3.13a [2.13–4.58]** | 4.63 [3.41–6.29]** | 4.54a [3.33–6.18]** |
| During labour/birth | 1.07 [0.77–1.49] | 1.08a [0.77–1.50] | 1.88 [1.24–2.84]** | 1.87a [1.23–2.83]* | 2.51 [1.81–3.47]** | 2.52a [1.82–3.49]** |
| During postpartum care in hospital | 1.00 [0.72–1.37] | 1.01a [0.73–1.39] | 2.14 [1.48–3.09]** | 2.15a [1.48–3.12]** | 1.99 [1.49–2.68]** | 2.06a [1.53–2.77]** |
| During postpartum care after going home† | 0.86 [0.62–1.19] | 0.85a [0.62–1.18] | 2.36 [1.60–3.47]** | 2.31a [1.56–3.41]** | 1.38 [1.02–1.86]* | 1.39a [1.03–1.88]* |
| **Care providers used language women could understand all of the time** | | | | | | |
| During pregnancy | 1.05 [0.76–1.47] | 1.06a [0.76–1.48] | 3.00 [1.92–4.70]** | 2.96a [1.88–4.64]** | 2.87 [2.06–3.99]** | 2.88a [2.06–4.01]** |
| During labour/birth | 1.05 [0.74–1.49] | 1.06a [0.74–1.51] | 2.31 [1.45–3.68]** | 2.27a [1.42–3.63]** | 2.38 [1.67–3.37]** | 2.40a [1.68–3.41]** |
| During postpartum care in hospital | 1.04 [0.76–1.43] | 1.06a [0.77–1.47] | 2.01 [1.36–2.96]** | 2.02a [1.37–3.00]** | 2.13 [1.58–2.89]** | 2.20a[1.62–2.99]** |
| During postpartum care after going home† | 1.10 [0.78–1.56] | 1.11a [0.78–1.58] | 2.22 [1.41–3.48]** | 2.17a [1.37–3.42]** | 1.24 [0.90–1.72] | 1.24a [0.89–1.73] |
| **Care providers treated women with respect all of the time** | | | | | | |
| During pregnancy | 1.24 [0.88–1.77] | 1.25a [0.88–1.79] | 2.88 [1.77–4.68]** | 2.82a [1.73–4.59]** | 3.85 [2.66–5.59]** | 3.84a [2.64–5.58]** |
| During labour/birth | 1.07 [0.75–1.54] | 1.08a[0.75–1.56] | 2.63 [1.59–4.35]** | 2.60a[1.57–4.30]** | 2.89 [1.99–4.20]** | 2.90a [1.99–4.24]** |
| During postpartum care in hospital | 1.15 [0.83–1.59] | 1.18a [0.85–1.62] | 2.05 [1.38–3.03]** | 2.08a [1.39–3.09]** | 2.17 [1.60–2.95]** | 2.27a [1.67–3.10]** |
| During postpartum care after going home† | 1.13 [0.79–1.62] | 1.15a [0.80–1.65] | 2.44 [1.52–3.92]** | 2.43a [1.51–3.92]** | 1.30 [0.93–1.82] | 1.33a [0.94–1.86] |
| **Care providers talked to women with kindness and understanding all of the time** | | | | | | |
| During pregnancy | 1.49 [1.06–2.10]* | 1.51a [1.06–2.13]* | 3.22 [2.03–5.12]** | 3.19a[2.00–5.08]** | 3.56 [2.52–5.03]** | 3.55a [2.50–5.03]** |
| During labour/birth | 1.05 [0.74–1.51] | 1.06a [0.74–1.52] | 2.42 [1.49–3.92]** | 2.39a [1.47–3.90]** | 2.42 [1.69–3.47]** | 2.40a [1.67–3.45]** |
| During postpartum care in hospital | 1.14 [0.83–1.57] | 1.16a [0.84–1.60] | 2.01 [1.36–2.97]** | 2.02a [1.36–2.99]** | 2.04 [1.51–2.75]** | 2.10a [1.55–2.85]** |
| During postpartum care after going home† | 1.11 [0.78–1.60] | 1.12a [0.78–1.62] | 2.49 [1.48–3.84]** | 2.36a [1.46–3.81]** | 1.22 [0.87–1.70] | 1.23a [0.88–1.73] |
| **Care providers treated women as an individual all of the time** | | | | | | |

(*Continued*)

**Table 7.** (Continued)

| | GP Shared Care[1] | | Public Midwifery Continuity Care[1] | | Private Obstetric Care[1] | |
|---|---|---|---|---|---|---|
| | aOR[(i)] [99% CI] | aOR[(ii)] [99% CI] | aOR[(i)] [99% CI] | aOR[(ii)] [99% CI] | aOR[(i)] [99% CI] | aOR[(ii)] [99% CI] |
| During pregnancy | 1.13 [0.81–1.58] | 1.13 [0.81–1.59] | 2.78 [1.79–4.35]** | 2.72[a] [1.73–4.25]** | 3.13 [2.24–4.38]** | 3.10[a] [2.21–4.34]** |
| During labour/birth | 0.97 [0.67–1.39] | 0.98[a][0.68–1.41] | 2.17 [1.34–3.52]** | 2.14[a] [1.32–3.49]** | 2.05 [1.43–2.94]** | 2.06[a] [1.43–2.97]** |
| During postpartum care in hospital | 1.10 [0.80–1.51] | 1.11[a] [0.81–1.53] | 2.08 [1.40–3.07]** | 2.07[a] [1.40–3.07]** | 2.16 [1.60–2.92]** | 2.18[a] [1.61–2.96]** |
| During postpartum care after going home† | 1.14 [0.80–1.63] | 1.14[a] [0.80–1.64] | 2.36 [1.48–3.76]** | 2.34[a] [1.46–3.75]** | 1.25 [0.90–1.75] | 1.27[a] [0.90–1.77] |
| Care providers were open and honest all of the time | | | | | | |
| During pregnancy | 1.20 [0.85–1.69] | 1.20[a] [0.85–1.70] | 3.23 [1.98–5.25]** | 3.14[a] [1.93–5.13]** | 3.00 [2.11–4.26]** | 2.96[a] [2.08–4.22]** |
| During labour/birth | 1.12 [0.78–1.60] | 1.13[a] [0.78–1.63] | 2.67 [1.63–4.38]** | 2.67[a] [1.62–4.39]** | 2.18 [1.53–3.11]** | 2.21[a] [1.55–3.17]** |
| During postpartum care in hospital | 1.11 [0.80–1.53] | 1.12[a] [0.81–1.55] | 2.00 [1.34–2.97]** | 1.97[a] [1.32–2.94]** | 2.06 [1.51–2.80]** | 2.08[a] [1.52–2.84]** |
| During postpartum care after going home† | 1.07 [0.74–1.53] | 1.06[a] [0.74–1.53] | 2.53 [1.55–4.12]** | 2.47[a] [1.51–4.03]** | 1.28 [0.91–1.80] | 1.28[a] [0.91–1.80] |
| Care providers respected women's privacy all of the time | | | | | | |
| During pregnancy | 1.24 [085–1.82] | 1.27 [0.86–1.86] | 2.92 [1.70–5.01]** | 2.87[a] [1.67–4.94]** | 3.49 [2.34–5.22]** | 3.54[a] [2.36–5.31]** |
| During labour/birth | 1.07 [0.74–1.56] | 1.08[a] [0.74–1.57] | 2.17 [1.33–3.55]** | 2.14[a] [1.31–3.52]** | 2.32 [1.60–3.36]** | 2.31[a] [1.59–3.36]** |
| During postpartum care in hospital | 1.09 [0.79–1.50] | 1.10[a] [0.80–1.52] | 2.18 [1.46–3.25]** | 2.15[a] [1.43–3.21]** | 2.33 [1.71–3.17]** | 2.37[a] [1.74–3.24]** |
| During postpartum care after going home† | 1.03 [0.71–1.48] | 1.02[a] [0.71–1.48] | 2.39 [1.45–3.93]** | 2.33[a] [1.41–3.84]** | 1.30 [0.92–1.84] | 1.30[a] [0.91–1.84] |
| Care providers respected women's decisions all of the time | | | | | | |
| During pregnancy | 1.45 [1.03–2.05]* | 1.45[a] [1.03–2.05]* | 2.93 [1.86–4.62]** | 2.84[a] [1.80–4.49]** | 2.99 [2.13–4.18]** | 2.91[a] [2.07–4.09]** |
| During labour/birth | 1.02 [0.71–1.45] | 1.01[a] [0.71–1.45] | 1.85 [1.83–2.91]** | 1.81[a] [1.15–2.86]** | 1.96 [1.38–2.77]** | 1.93[a] [1.36–2.75]** |
| During postpartum care in hospital | 1.01 [0.74–1.39] | 1.01[a] [0.73–1.39] | 1.86 [1.26–2.73]** | 1.82[a] [1.23–2.68]** | 1.98 [1.46–2.67]** | 2.00[a] [1.48–2.71]** |
| During postpartum care after going home† | 1.08 [0.76–1.53] | 1.09[a] [0.76–1.55] | 2.13 [1.35–3.35]** | 2.10[a] [1.33–3.32]** | 1.23 [0.88–1.71] | 1.24[a] [0.89–1.74] |
| Care providers genuinely cared about women's wellbeing all of the time | | | | | | |
| During pregnancy | 1.24 [0.88–1.74] | 1.24[a] [0.88–1.75] | 3.26 [2.03–5.22]** | 3.18[a] [1.98–5.12]** | 3.82 [2.69–5.42]** | 3.78[a] [2.65–5.38]** |
| During labour/birth | 1.02 [0.71–1.48] | 1.03[a] [0.71–1.49] | 2.09 [1.29–3.40]** | 2.06[a] [1.26–3.36]** | 2.82 [1.93–4.12]** | 2.80[a] [1.91–4.10]** |
| During postpartum care in hospital | 1.09 [0.79–1.49] | 1.01[a] [0.80–1.52] | 1.93 [1.31–2.86]** | 1.92[a] [1.30–2.85]** | 2.15 [1.59–2.91]** | 2.21[a] [1.63–3.01]** |
| During postpartum care after going home† | 1.00 [0.70–1.43] | 1.01[a] [0.70–1.45] | 2.12 [1.33–3.39]** | 2.09[a] [1.30–3.36]** | 1.14 [0.82–1.60] | 1.16[a][0.82–1.62] |
| Women were confident in the skills of care providers all of the time | | | | | | |
| During pregnancy | 1.02 [0.74–1.39] | 1.00[a] [0.73–1.38] | 2.45 [1.67–3.06]** | 2.37[a] [1.61–3.48]** | 5.96 [4.32–8.24]** | 5.83[a] [4.21–8.07]** |

(*Continued*)

**Table 7.** (Continued)

| | GP Shared Care[1] | | Public Midwifery Continuity Care[1] | | Private Obstetric Care[1] | |
|---|---|---|---|---|---|---|
| | aOR[(i)] [99% CI] | aOR[(ii)] [99% CI] | aOR[(i)] [99% CI] | aOR[(ii)] [99% CI] | aOR[(i)] [99% CI] | aOR[(ii)] [99% CI] |
| During labour/birth | 1.02 [0.73–1.41] | 1.03[a] [0.74–1.44] | 1.81 [1.21–2.73]** | 1.82[a] [1.21–2.75]** | 2.88 [2.07–3.99]** | 2.91[a] [2.09–4.05]** |
| During postpartum care in hospital | 0.88 [0.64–1.20] | 0.89[a] [0.65–1.22] | 1.69 [1.17–2.44]** | 1.70[a][1.17–2.46]** | 1.54 [1.15–2.06]** | 1.59[a] [1.18–2.13]** |
| During postpartum care after going home† | 1.10 [0.79–1.51] | 1.09[a] [0.78–1.51] | 2.16 [1.46–3.20]** | 2.09[a] [1.41–3.11]** | 1.44 [1.06–1.94]* | 1.46[a][1.07–1.98]* |
| Women knew what was happening all of the time | | | | | | |
| During pregnancy | 1.31 [0.96–1.80] | 1.30[a] [0.94–1.79] | 2.47 [1.70–3.58]** | 2.39[a] [1.64–3.48]** | 3.41 [2.53–4.60]** | 3.48[a][2.57–4.71] v |
| During labour/birth | 1.10 [0.81–1.51] | 1.12[a] [0.82–1.54] | 1.62 [1.13–2.34]** | 1.64[a] [1.13–2.37]** | 1.87 [1.40–2.49]** | 1.91[a] [1.44–2.59]** |
| During postpartum care in hospital | 0.90 [0.65–1.25] | 0.91[a][0.65–1.26] | 1.76 [1.22–2.54]** | 1.76[a] [1.22–2.55]** | 1.82 [1.36–2.45]** | 1.88[a] [1.39–2.53]** |
| During postpartum care after going home† | 0.91 [0.66–1.26] | 0.91[a] [0.65–1.26] | 1.54 [1.05–2.27]* | 1.51[a] [1.02–2.23]* | 1.09 [0.81–1.47] | 1.13[a] [0.83–1.53] |
| Women felt comfortable asking questions all of the time | | | | | | |
| During pregnancy | 1.13 [0.82–1.56] | 1.12[a] [0.81–1.55] | 2.75 [1.81–4.19]** | 2.67[a] [1.75–4.07]** | 2.60 [1.90–3.56]** | 2.57[a] [1.87–3.54]** |
| During labour/birth | 1.19 [0.86–1.65] | 1.21[a] [0.87–1.68] | 1.81 [1.21–2.70]** | 1.80[a] [1.20–2.69]** | 2.12 [1.55–2.90]** | 2.17[a][1.58–2.98]** |
| During postpartum care in hospital | 0.98 [0.72–1.35] | 1.00[a][0.73–1.38] | 1.80 [1.24–2.61]** | 1.82[a] [1.25–2.64]** | 1.63 [1.22–2.19]** | 1.70[a] [1.26–2.78]** |
| During postpartum care after going home† | 0.96 [0.68–1.34] | 0.94[a] [0.67–1.32] | 1.95 [1.28–2.98]** | 1.88[a] [1.23–2.89]** | 1.17 [0.85–1.60] | 1.17[a] [0.85–1.62] |
| Women felt in control all of the time | | | | | | |
| During pregnancy | 1.34 [0.97–1.84] | 1.31[a] [0.94–1.81] | 2.71 [1.87–3.93]** | 2.59[a] [1.78–3.78]** | 2.67 [1.98–3.59]** | 2.65[a] [1.95–3.57]** |
| During labour/birth | 1.33 [0.96–1.85] | 1.37[a] [0.98–1.91] | 2.02 [1.39–2.93]** | 2.05[a] [1.41–3.00]** | 2.01 [1.48–2.71]** | 2.13[a] [1.57–2.91]** |
| During postpartum care in hospital | 1.07 [0.77–1.48] | 1.09[a][0.78–1.52] | 2.23 [1.54–3.23]** | 2.28[a] [1.56–3.34]** | 2.03 [1.50–2.74]** | 2.19[a] [1.61–2.98]** |
| During postpartum care after going home† | 1.01 [0.73–1.39] | 1.02[a] [0.73–1.41] | 1.67 [1.14–2.44]** | 1.64[a] [1.11–2.43]* | 1.13 [0.84–1.53] | 1.18[a] [0.87–1.61] |
| Women never received conflicting information/advice from different care providers | | | | | | |
| During pregnancy | 1.09 [0.79–1.51] | 1.07[a] [0.77–1.49] | 1.64 [1.14–2.37]** | 1.60[a] [1.10–2.31]* | 2.87 [2.13–3.87]** | 2.94[a] [2.17–3.98]** |
| During labour/birth | 1.18 [0.85–1.64] | 1.19[a] [0.86–1.66] | 1.60 [1.08–2.37]* | 1.61[a] [1.09–2.39]* | 1.77 [1.30–2.41]** | 1.80[a][1.32–2.46]** |
| During postpartum care in hospital | 1.09 [0.79–1.51] | 1.12[a] [0.81–1.56] | 1.45 [1.01–2.10]* | 1.51[a] [1.04–2.19]* | 0.98 [0.73–1.32] | 1.06[a] [0.78–1.43] |
| During postpartum care after going home† | 1.12 [0.81–1.55] | 1.16[a] [0.83–1.61] | 1.74 [1.20–2.53]** | 1.78[a] [1.21–2.60]** | 1.12 [0.83–1.51] | 1.21[a] [0.89–1.65] |
| Women felt safe all of the time | | | | | | |
| During pregnancy | 1.04 [0.75–1.44] | 1.03[a] [0.74–1.44] | 2.42 [1.57–3.71]** | 2.32[a] [1.51–3.58]** | 2.92 [2.11–4.04]** | 2.91[a] [2.09–4.04]** |
| During labour/birth | 1.02 [0.72–1.40] | 1.02[a] [0.73–1.43] | 1.86 [1.23–2.82]** | 1.88[a] [1.24–2.86]** | 2.22 [1.61–3.06]** | 2.29[a] [1.65–3.17]** |
| During postpartum care in hospital | 0.92 [0.67–1.28] | 0.92[a] [0.66–1.28] | 1.92 [1.28–2.88]** | 1.85[a] [1.23–2.79]** | 2.14 [1.56–2.93]** | 2.16[a] [1.57–2.97]** |

(Continued)

**Table 7.** (Continued)

| | GP Shared Care[1] | | Public Midwifery Continuity Care[1] | | Private Obstetric Care[1] | |
|---|---|---|---|---|---|---|
| | aOR(i) [99% CI] | aOR(ii) [99% CI] | aOR(i) [99% CI] | aOR(ii) [99% CI] | aOR(i) [99% CI] | aOR(ii) [99% CI] |
| During postpartum care after going home† | 1.02 [0.72–1.45] | 1.02[a] [0.71–1.45] | 2.04 [1.31–3.18]** | 1.99[a] [1.27–3.12]** | 1.27 [0.91–1.76] | 1.28[a] [0.92–1.79] |
| Women never wanted to be more involved in decisions | | | | | | |
| During pregnancy | 1.17 [0.85–1.61] | 1.15[a] [0.83–1.58] | 1.84 [1.27–2.66]** | 1.75[a] [1.21–2.53]** | 2.21 [1.65–2.97]** | 2.13[a] [1.59–2.87]** |
| During labour/birth | 0.98 [0.71–1.34] | 0.97[a] [0.71–1.34] | 1.20 [0.83–1.73] | 1.18[a] [0.82–1.71] | 1.65 [1.23–2.22]** | 1.64[a] [1.22–2.20]** |
| During postpartum care in hospital | 1.20 [0.87–1.64] | 1.20[a] [0.87–1.66] | 1.55 [1.07–2.23]* | 1.52 [1.05–2.20]* | 1.91 [1.43–2.56]** | 1.94 [1.44–2.60]** |
| During postpartum care after going home† | 1.11 [0.80–1.54] | 1.10[a] [0.79–1.52] | 1.31 [0.90–1.92] | 1.26[a] [0.86–1.85] | 1.34 [0.99–1.81] | 1.31[a] [0.97–1.79] |
| Women felt care providers were on their side all of the time | | | | | | |
| During pregnancy | 1.37 [0.99–1.88] | 1.35 [0.98–1.87] | 3.04 [2.02–4.59]** | 2.93[a] [1.94–4.42]** | 4.09 [2.98–5.63]** | 4.00[a] [2.90–5.52]** |
| During labour/birth | 1.24 [0.89–1.74] | 1.25[a] [0.89–1.76] | 1.72 [1.14–2.60]** | 1.73[a] [1.14–2.62]** | 2.66 [1.91–3.71]** | 2.67[a] [1.91–3.74]** |
| During postpartum care in hospital | 0.99 [0.72–1.35] | 1.01[a][0.73–1.38] | 1.86 [1.28–2.69]** | 1.88[a] [1.29–2.73]** | 1.73 [1.29–2.31]** | 1.79[a] [1.33–2.41]** |
| During postpartum care after going home† | 1.13 [0.81–1.58] | 1.13[a] [0.81–1.58] | 1.97 [1.31–2.97]** | 1.92[a] [1.27–2.91]** | 1.36 [0.99–1.86] | 1.40[a] [1.02–1.92]* |
| Women never wished care providers had more time to talk | | | | | | |
| During pregnancy | 1.14 [0.82–1.57] | 1.12[a] [0.81–1.56] | 2.34 [1.62–3.39]** | 2.26[a] [1.56–3.29]** | 2.43 [1.81–3.28]** | 2.40[a] [1.77–3.24]** |
| During labour/birth | 1.28 [0.93–1.75] | 1.31[a] [0.95–1.80] | 1.96 [1.35–2.85]** | 1.99[a] [1.37–2.91]** | 1.72 [1.29–2.31]** | 1.77[a][1.32–2.37]** |
| During postpartum care in hospital | 1.02 [0.73–1.42] | 1.02[a] [0.73–1.43] | 1.95 [1.34–2.83]** | 1.95[a] [1.34–2.84]** | 1.81 [1.34–2.45]** | 1.85[a][1.36–2.51]** |
| During postpartum care after going home† | 1.00 [0.72–1.38] | 1.01[a] [0.73–1.39] | 1.55 [1.07–2.24]* | 1.53 [1.05–2.22]* | 1.35 [1.00–1.81] | 1.39 [1.02–1.87]* |
| Women were very well looked after by care providers | | | | | | |
| During pregnancy | 1.06 [0.77–1.45] | 1.06[a] [0.77–1.45] | 3.11 [2.07–4.66]** | 3.04[a] [2.02–4.57]** | 4.24 [3.09–5.82]** | 4.21[a] [3.06–5.80]** |
| During labour and birth | 1.27 [0.91–1.76] | 1.28[a] [0.92–1.79] | 2.05 [1.36–3.09]** | 2.08[a] [1.38–3.15]** | 2.98 [2.15–4.14]** | 3.03[a][2.18–4.22]** |
| During postpartum care in hospital | 1.03 [0.75–1.41] | 1.04[a] [0.76–1.44] | 1.84 [1.28–2.66]** | 1.86[a][1.29–2.70]** | 2.13 [1.59–2.86]** | 2.17[a][1.61–2.92]** |
| During postpartum care after going home† | 0.89 [0.65–1.23] | 0.91[a][0.66–1.25] | 1.88 [1.28–2.70]** | 1.88[a] [1.28–2.77]** | 0.78 [0.58–1.04] | 0.80[a] [0.59–1.08] |

*Note*: Adjustments for sociodemographic characteristics and reproductive history include: maternal age, BMI, area of residence, education, language spoken at home, country of birth and previous caesarean; aOR = adjusted odds ratio.

[1] vs. Standard Public Care

[a] adjusted for sociodemographic characteristics, reproductive history, parity and complications during index pregnancy (depression, gestational diabetes, amount of amniotic fluid was a concern, baby was too small).

* $p < .01$

** $p < .001$

† Of the women who received postpartum care at home ($n = 2637$)

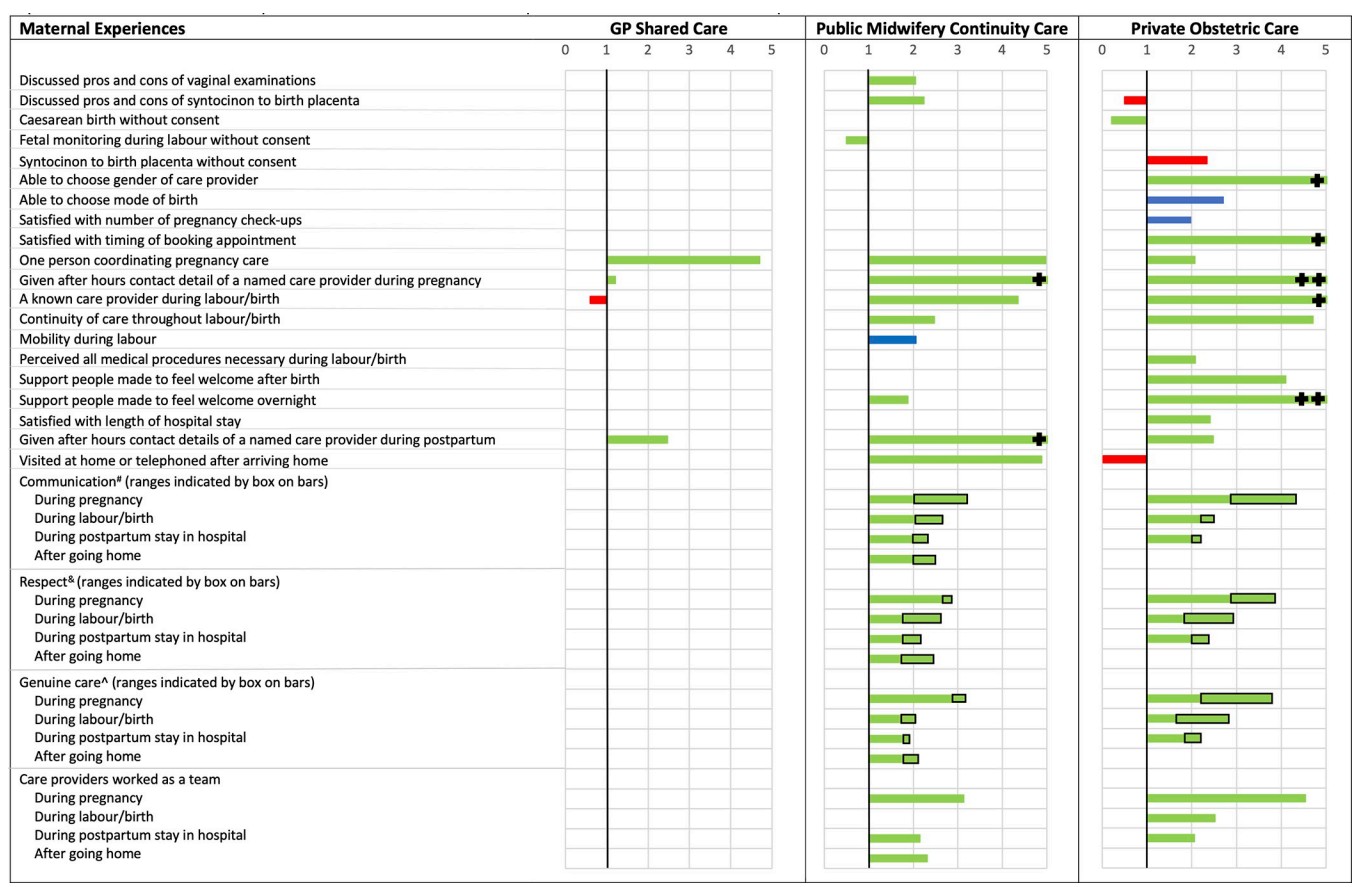

**Fig 2. Adjusted odds[*][#] of maternal experiences significantly different to Standard Public Care[∞].** Green indicates more optimal experiences, red indicates less optimal experiences and blue indicates experiences with differential value dependent on consumer needs and preferences.

result likely reflects the change in primary care provider between antenatal and intrapartum care for women who receive care in GP Shared Care models.

A main point of difference between the compared MMCs was mode of birth. Women in Public Midwifery Continuity Care were more likely to have an unassisted vaginal birth and less likely to have a scheduled caesarean birth or an assisted vaginal birth. Women in the Private Obstetric model were more likely to have a scheduled caesarean birth and less likely to have an unassisted vaginal birth or an unscheduled caesarean birth. These results are comparable to previous studies comparing mode of birth between MMCs, with consistently higher rates of unassisted vaginal births and lower rates of caesarean births observed in Public Midwifery Continuity models [8–11, 13, 19] and higher rates of elective caesarean births in Private Obstetric Care [19]. Variations in the rate of different modes of birth may be representative of women's and clinicians' birth philosophies about childbirth as a medical procedure and practice standards between MMCs, as well as women's access to mode of birth choices in some MMCs. In this study, women in Private Obstetric Care were more likely to report that they were able to choose their mode of birth than women in Standard Public Care, whereas women in Public Midwifery Continuity model were less likely to report being able to choose. This may have contributed to differences in the rate of scheduled caesareans. Women who elect for a caesarean birth might be more likely to access Private Obstetric Care and less likely to access Public Midwifery Continuity Care. Further comparisons between MMCs on the basis of

women's perceived access to their preferred mode of birth and likelihood of having the type of birth they wanted may be more helpful for women's MMC decision-making.

Differences in rates of skin-to-skin contact between MMCs may have been associated with differences in mode of birth. Our findings that skin-to-skin contact the first time women held their baby was more likely in Public Midwifery Continuity Care and less likely in Private Obstetric Care compared to Standard Public Care, were not sustained after adjustment for mode of birth. Women who have a caesarean birth, which was significantly more likely in Private Obstetric Care and less likely in Public Midwifery Continuity Care than in Standard Public Care, have been found to be less likely to experience immediate skin-to-skin contact with their baby [31–33]. Ours is the first study to compare rates of skin-to-skin contact between MMCs in Australia [5], although it has been cited by women as an important factor for their MMC decision-making [7]. Further research on the interactions between MMC, mode of birth, and skin-to-skin contact could better inform women of the interrelationships between variations in outcomes and experiences associated with alternative MMCs.

We found comparably decreased odds of NICU admission for both Public Midwifery Continuity Models and Private Obstetric Models compared to Standard Public Care, and decreased odds of preterm birth for Public Midwifery Continuity Care, even after accounting for relevant maternal and clinical characteristics. Trials conducted internationally have found similar (though smaller on average) reduced likelihood of preterm birth for women who receive Midwifery Continuity Care than those in other MMCs [16]. NICU admission has been found to be lower in a Public Midwifery Continuity Model than in either Standard Public or Private Obstetric Care for women receiving care in one hospital in Australia [19], but not significantly different between Midwifery-led care and other models in an international review and meta-analysis [17]. Notably, we were unable to compare MMCs on infant survival outcomes in this study because our sampling strategy excluded women who had a stillbirth or a neonatal death from our standard data collection procedures, although lower risk of fetal and infant loss have been found for Midwifery Continuity Care compared to other MMCs in the international literature [16].

Discussing the pros and cons of pregnancy procedures and experiencing procedures without consent during pregnancy did not differ between MMCs. However, women in Public Midwifery Continuity Care had higher odds of having their care providers discuss the pros and cons for several intrapartum procedures (induction of labour, vaginal examinations, fetal monitoring and receiving Syntocinon to birth their placenta), consistent with findings from one other study that compared information provision between Public Midwifery Continuity Care and Standard Public Care in Australia [21]. They were also less likely to experience vaginal examinations and fetal monitoring without their consent. Women in Private Obstetric Care were more likely to have their care providers discuss the pros and cons of caesarean birth with them and less likely to experience a caesarean birth without consent. These results indicate that women in Public Midwifery Continuity and Private Obstetric models were more likely than those in Standard Public Care to experience informed consent for specific, and different, medical interventions. Research in Canada has shown that women's overall experience of informed consent processes differs by model of care (defined by the primary care provider) [34], but ours is the first research globally to compare procedure-specific information provision and consent by MMC. In 2020, the Queensland State Government Health Department released guidelines specifying the need to discuss and document the potential benefits and risks of recommended care and alternatives when a pregnant woman declines, or expresses an intention to decline, aspects of recommended maternity care [35]. The guidelines, for use in Queensland public hospitals where Standard Public Care is delivered, may have since affected the differences in informed consent processes between MMCs reported here, although the

impact of implementing those guidelines on women's experiences of informed consent has not been evaluated. Importantly, guidance on the informed decision-making processes which may precede women's refusal of aspects of recommended maternity care is not provided. Further research is needed to confirm whether the differences between MMCs in Queensland reported here have been sustained since the time of data collection.

Women in GP Shared Care, Public Midwifery Continuity Care and Private Obstetric Care were more likely to have one person coordinating their pregnancy care. Continuity of carer and having a known care provider during labour and birth were more likely in Public Midwifery Continuity and Private Obstetric models. This was unsurprising, given that both models are structured for women to receive care from the same midwife, team of midwives or named obstetrician across antepartum and intrapartum care [5]. However, contact from a nurse or midwife at home after birth was more likely in Public Midwifery Continuity Care and less likely in Private Obstetric Care. Differences in postnatal contact at home may be somewhat related to our other findings for a difference in mean length of maternal hospital stay between MMCs [36]. Women in Private Obstetric Care had a mean increase of 36 hours of postpartum hospital stay relative to Standard Public Care, whereas hospital discharge was a mean 16 hours earlier in Public Midwifery Continuity Care. Domiciliary care is provided less frequently by private hospitals than public hospitals in the Australian Health Care system [36, 37]. Women with access to both private and public MMCs should be made aware of differences in the provision of postpartum at-home care in making MMC decisions, given that women who birth in private facilities are more likely to spontaneously report concerns about their level of care after hospital discharge than women who birthed in public facilities in Queensland [38].

Women in Public Midwifery Continuity Care consistently experienced better interpersonal and overall quality of care across all care stages. Women in the Private Obstetric model experienced better interpersonal and overall quality of care across antenatal, intrapartum, and postpartum care in hospital. Findings from other studies in Australia [24, 39, 40] suggest continuity of care is associated with higher care satisfaction and better postpartum health outcomes. Increased length of hospital stay has been positively associated with satisfaction of postnatal care in hospital [39], which may explain why women in the Private Obstetric model reported better experiences of care during this period, despite not rating their interpersonal and overall quality of care at home better than Standard Public Care. Moreover, evidence from a Western Australian study suggests that women who receive maternity care in the private sector rate aspects of their postnatal care at home less favourably than women who birthed in the public sector [41]. Experiences of interpersonal care were comparable between GP Shared and Standard Public Care.

Using women's self-reported data from a population sample we were able to provide the first direct comparison of four MMCs in Australia. This is the first study to compare GP Shared Care with other available models (separately from Standard Public Care), despite GP Shared Care being the most frequently discussed model in referral conversations between women and GPs [4]. Ours is only the second study to include a comparison across publicly and privately funded models. Existing evidence has been limited by comparisons of only two to three models within the public care system, limiting the usefulness for women's MMC decision-making. Measuring MMC using self-reported data from recipients is more closely aligned with the aspects of care actually experienced by women (rather than their 'clinical' allocation) and is less affected by differences in how MMCs are defined or operationalised across birth facilities and geographical locations [20]. However, our algorithm for categorising women according to a specific MMC did not account for transitions between models (since women were classified as having received a single MMC) and was biased towards classification of the final model received (i.e., for intrapartum care). Importantly for decision-making about MMC

allocation and referral in early pregnancy, this means our analyses is based on 'treatment received' rather than 'intention-to-treat'. We attempted to reduce this bias as much as possible by adjusting analyses for clinical characteristics that may affect transitions between models from initial allocation to time of birth. A further limitation associated with our MMC coding algorithm was failing to distinguish between Caseload and other Midwifery Continuity models, which have been found to differ in clinical outcomes in other settings (e.g., [42]). Nevertheless, using women's self-reported data allowed us to compare both clinical outcomes and experiential measures with previously unmatched breadth. Women's self-reported data about their maternity care is at least as accurate as medical records [43–45]. Experiential outcomes have received less research attention despite having been identified as important for women's informed decision-making [7].

Our use of population data collected from women who birthed across numerous hospitals in Queensland is less affected by variations in how MMCs are defined and operationalised across different facilities and regions, a limitation of previous studies who have sampled from only one or few hospitals or birthing facilities. However, our response rate to the survey was 30.4% and the sample did underrepresent younger women, women from remote or rural areas, women born outside of Australia, multiparous women, and Aboriginal and Torres Strait Islander women in comparison to the total population of birthing women in Queensland [29, 30]. Maternal and infant outcomes and access to maternity care services in Australia can differ for women in these groups [46–52]. Although we adjusted for sociodemographic variables and reproductive history, future comparisons of outcomes and experiences within these specific populations of women may reduce selection bias. We were unable to adjust for Indigenous identification due to low frequencies of Aboriginal and Torres Strait Islander women in our sample. Disparities in maternal and infant health have persisted between Aboriginal and Torres Strait Islander women and babies and the non-indigenous birthing population in Australia, with Aboriginal and Torres Strait Islander women experiencing disproportionately higher rates of adverse maternal and infant health outcomes [46, 47]. To ensure Aboriginal and Torre Strait Islander women have access to adequate information regarding MMCs that is encompassing of their preferences, values, and needs, future comparisons of Aboriginal and Torres Strait Islander women's experiences of alternative MMCs is needed [53].

A limitation of our work for usefully informing decisions about MMC arises from our inability to have included Private Midwifery Care in our comparisons due to low frequencies of women who received care in this model. At the time of data collection in 2012, significant Australian Federal government reforms to improve access to Private Midwifery Care had recently been enacted. In 2010, Australia introduced public healthcare funding for care delivered by Medicare eligible Private Practice Midwives, which included Medicare rebates for antenatal and postpartum care delivered by eligible Private Practice Midwives alongside mechanisms for collaborative arrangements between eligible Private Practice Midwives and hospitals to provide 'visiting access' for in-hospital intrapartum care with continuity of care from the woman's chosen private midwife. Queensland, where the current study was conducted, utilised the national reforms to provide visiting access to Medicare eligible private practice midwives "earlier and more fully" than other Australian states and territories [54]. The most recent data suggest that Private Midwifery Models account for 2.2% of all MMCs available in Australia and 5% of all MMCs in Queensland [55], although how this compares to earlier pre-reform availability and the proportion of women who use Private Midwifery Care remains unknown. Intrapartum care at home (i.e., homebirth) has remained un-funded in the national (public) health insurance system (i.e., all costs are privately funded by the woman) [56] and the percentage of women who give birth at home in Queensland has only marginally increased from 0.1% to 0.2% in the period between 2012 and 2019 [46]. There is limited evidence on the

outcomes of Private Midwifery Care in Australia. Two studies in Queensland compared maternal and neonatal clinical outcomes for women who received care from a specific Private Midwifery service [57] and from Private Midwives with visiting access to a specific hospital [54] respectively, against National Core Maternity Indicators. Women who received Private Midwifery Care had higher rates of spontaneous labour onset, received less pharmacological pain relief during labour, had more spontaneous vaginal births and fewer caesarean births, were more likely to have an intact perineum and received fewer episiotomies in comparison to national rates [54]. Women in Private Midwifery Care also had fewer preterm births and were less likely to have their infants admitted to the NICU [54]. There is no existing evidence providing direct comparisons between Private Midwifery Care and other MMCs in Australia [5]. More evidence on women's experiences in Private Midwifery Models in ways that allow direct comparison and include experiential measures of care is needed.

We were unable to compare out-of-pocket costs between MMCs in this research because it was not assessed in the survey. However, women cite it as one of the most important factors to their decision-making [7]. An assessment of total economic costs, including private out-of-pocket costs, and health outcomes presented in a cost-effectiveness analysis may be an important factor to inform policy decision-making about the provision of alternative models. As in all policy decision-making, multiple criteria should be considered in decisions that create the right mix of MMCs in a market for maternity care. In this study, we have contributed critical information on clinical outcomes and experiences. However, these should be weighed alongside the need to be responsive to women's preferences for MMCs and the costs of doing so. Previous studies examining costs between MMCs in Australia have focused on the cost per woman from the hospital perspective and have indicated that Public Midwifery Continuity Care has a lower mean cost per woman in comparison to both Standard Public Care and Private Obstetric Care [19]. Such comparisons are less useful for women's MMC decision-making than the prospective out-of-pocket costs for women associated with each MMC. Comparisons of publicly and privately funded maternity care indicate out-of-pocket costs up to eight times higher for women in the Private sector [58]. Further comparisons of out-of-pocket costs associated with different publicly funded models is warranted.

Presenting odds of outcomes and experiences in alternative MMCs with a consistent referent of Standard Public Care was performed to compare alternatives to the universally accessible MMC, but this may not be the most useful comparison for all women's decision-making. A focus on comparisons of MMCs that respond best to women's self-reported decision-making needs is critical and requires a broad assessment of traditional clinical outcomes alongside experiential measures [7]. Women have different values and preferences for their maternity care. What is considered more or less important will vary between women [7]. Data like this could be transformed into an adaptive decision aid that allows women to choose comparisons between outcomes and experiences they are most interested in for their personal decision-making about MMC.

## Conclusion

There are major variations between MMCs in the likelihood of outcomes and experiences for maternity care consumers, and particularly for models of care that are less frequently discussed with women in MMC decision-making. Usable information about the prospective clinical outcomes and experiences associated with each MMC can be provided during the early antenatal period to enable informed MMC decision-making that reflects each woman's individual maternity care needs and preferences. Most Australian women visit a GP as their first antenatal care providers but very few are informed of all available MMCs during their first antenatal

care visits. Translating evidence into usable decision tools should be a priority to address known limitations in the accessibility and quality of decision-making about MMC. There remains a need to routinely assess the full range of outcomes and experiences that are important for decision-making at the population level, including out-of-pocket costs to women. Sufficiently engaging women who receive Private Midwifery Care in comparable evaluation of their outcomes and experiences should be prioritised.

## Supporting information

**S1 Table. Measurement of outcomes and potential confounders.**
(DOCX)

**S2 Table. Frequencies and crude odds ratios for obstetric interventions and maternal and infant clinical outcomes by model of care.**
(DOCX)

**S3 Table. Frequencies and crude odds ratio for information provision and decision-making outcomes by model of care.**
(DOCX)

**S4 Table. Frequencies and crude odd ratios for maternal experiences during pregnancy, labour/birth, and postpartum care by model of care.**
(DOCX)

**S5 Table. Frequencies and crude odds ratios of maternal experiences of interpersonal and overall quality of care by maternity model of care.**
(DOCX)

## Acknowledgments

We are grateful to all women who provided information about their maternity care experience via the Having a Baby in Queensland Survey, 2012. Dr Gabrielle Stevens created the coding schema for categorising maternity model of care in the data set.

## Author Contributions

**Conceptualization:** Yvette D. Miller.

**Data curation:** Yvette D. Miller, Jessica Tone.

**Investigation:** Yvette D. Miller.

**Methodology:** Yvette D. Miller, Jessica Tone, Sutapa Talukdar, Elizabeth Martin.

**Supervision:** Yvette D. Miller.

**Validation:** Yvette D. Miller, Elizabeth Martin.

**Visualization:** Jessica Tone.

**Writing – original draft:** Yvette D. Miller, Jessica Tone.

**Writing – review & editing:** Yvette D. Miller, Jessica Tone, Sutapa Talukdar, Elizabeth Martin.

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
