## [Decision Letter · Decision Letter 0]

4 Mar 2022

PONE-D-22-04372A direct comparison of patient-reported outcomes and experiences in alternative models of maternity care in Queensland, Australia.PLOS ONE

Dear Dr. Miller,

Thank you for submitting your manuscript to PLOS ONE. After careful consideration, we feel that it has merit but does not fully meet PLOS ONE’s publication criteria as it currently stands. Therefore, we invite you to submit a revised version of the manuscript that addresses the points raised during the review process. Please submit your revised manuscript by Apr 18 2022 11:59PM. If you will need more time than this to complete your revisions, please reply to this message or contact the journal office at plosone@plos.org. Please include the following items when submitting your revised manuscript:A rebuttal letter that responds to each point raised by the academic editor and reviewer(s). You should upload this letter as a separate file labeled 'Response to Reviewers'.A marked-up copy of your manuscript that highlights changes made to the original version. You should upload this as a separate file labeled 'Revised Manuscript with Track Changes'.An unmarked version of your revised paper without tracked changes. You should upload this as a separate file labeled 'Manuscript'.

We look forward to receiving your revised manuscript.

Kind regards,

Hannah Dahlen, RN, RM, BN (Hons), MCommN, PhD FACM

Academic Editor

PLOS ONE

Journal Requirements:

"The authors received no specific funding for this work. The data on which this article is based was collected as part of a statewide survey program funded by the Queensland Government, at The University of Queensland. The funders nor the recipients of that funding had a role in study design, data collection and analysis, decision to publish, or preparation of the manuscript."

"We have read the journal's policy and the authors of this manuscript have the following competing interests:YM has previously received funding for the development of patient decision aids, including resources for women to choose between models of maternity care, and for establishing a state-wide survey of recent maternity consumers’ experience of maternity care across different models of care in Queensland, Australia. The funding bodies for that work had no involvement in the research reported here. EM is employed by a health service at which some participants in this study gave birth. EM was not employed by the health service at the time of data collection (2012) and the health service had no involvement or influence in the analysis of the data for the work reported in this manuscript. JT and ST have no competing interests to declare."

5. We note you have included a table to which you do not refer in the text of your manuscript. Please ensure that you refer to Table 7 in your text; if accepted, production will need this reference to link the reader to the Table.

6. Please include a copy of Table 8 which you refer to in your text on page 37.

Reviewers' comments:

Reviewer's Responses to Questions

**Comments to the Author**

1. Is the manuscript technically sound, and do the data support the conclusions?

Reviewer #1: Yes

Reviewer #2: Yes

2. Has the statistical analysis been performed appropriately and rigorously? 

Reviewer #1: Yes

Reviewer #2: I Don't Know

3. Have the authors made all data underlying the findings in their manuscript fully available?

Reviewer #1: Yes

Reviewer #2: Yes

4. Is the manuscript presented in an intelligible fashion and written in standard English?

Reviewer #1: Yes

Reviewer #2: Yes

5. Review Comments to the Author

Reviewer #1: Abstracts have different wording. Please can they agree? The abstract in the paper is more informative.

Rationale for the study is well argued. The aim of this study was to directly compare women’s pregnancy, labour, birth, and postpartum outcomes and experiences across the major MMC categories offered in Queensland, Australia, using data collected in 2012 from a state-wide sample of women who had given birth in Queensland.

This paper conducts secondary analyses of self-reported data collected in 2012 from a state-wide sample of women who had recently given birth in Queensland Australia. Please add to abstract where research is conducted and when.

This retrospective cohort study was conducted using data obtained from the 2012 Having a Baby in Queensland Survey. Please add response rate to abstract.

This data is 10 years old. Please add to limitations and any relevant changes in policy or practice during this time.

Of 2,802 women, 18.2% received Standard Public Care, 21.7% received GP Shared Care, 12.9% received Public Midwifery Continuity Care, and 47.1% received Private Obstetric Care. Please specify differences between these models.

Logistic regression was used to estimate the odds of 34 outcomes and experiences associated with three models (GP Shared Care, Public Midwifery Continuity Care, Private Obstetric Care) compared with Standard Public Care, adjusting for relevant maternal characteristics and clinical covariates.

Line 73. Does private midwifery care provide continuity?

Line 93 Specify which antenatal care providers ?

Table 1 clarify who provides obstetric care if required in each model and who pays for this?

Line 239 Classification of MMC did not capture transitions between models of care during pregnancy. Where women transitioned between models of care during the index pregnancy, women were classified into the final model received in pregnancy to allow comparison of outcomes under a single model of care received. My understanding is that women are not classified by the model they started with, but the model they ended up in. Is this correct? If so, please justify as this is not intention to treat analysis and introduces bias?

Line 304 Low response rate. Comment upon limitations of study. Please add weblink to survey questions if available.

There are many many comparisons in this paper by differing models. Is it possible to produce a diagram/infographic summarising these to help the reader interpret the findings.

Line 618 Please report key infant outcomes here for all models as a balancing measure.

There is little reference to work in the filed in the discussion. What does this work add to this body of international literature?

Reviewer #2: Thank you for the opportunity to review this paper. I found the paper to be very well written and thorough. I preferred the tables in the supplementary that included p values and think this format would be better for the wider readership than the tables in the manuscript that only have OR presented as it helps the reader focus on the statistically significant outcomes. There is also little reference to the CI in the results but the tables include them.

My biggest concern is with the age of the dataset being over 10 years since collection. You haven't mentioned in the limitations the fact that the data is over 10 years old and that there has been development of the PPM and MGP model across Australia and in QLD during that time. When you collated this data the 2010 maternity care changes had just come into effect and there would have been very few endorsed (eligible) midwives at the time, now there are many more and now also with visiting rights, especially in QLD. I think this really needs to be explained in this paper. Otherwise you are suggesting this data is more current and relevant than it is.

6. PLOS authors have the option to publish the peer review history of their article (what does this mean?). If published, this will include your full peer review and any attached files.

Reviewer #1: **Yes: **jane sandall

Reviewer #2: No

---

## [Author Response · Author response to Decision Letter 0]

23 May 2022

All responses to specific reviewer and journal comments are outlined in the Cover Letter and Response to Reviewers.

---

## [Decision Letter · Decision Letter 1]

24 Jun 2022

A direct comparison of patient-reported outcomes and experiences in alternative models of maternity care in Queensland, Australia.

PONE-D-22-04372R1

Dear Dr. Miller,

We’re pleased to inform you that your manuscript has been judged scientifically suitable for publication and will be formally accepted for publication once it meets all outstanding technical requirements.

Kind regards,

Hannah Dahlen, RN, RM, BN (Hons), MCommN, PhD FACM

Academic Editor

PLOS ONE

Additional Editor Comments (optional):

Reviewers' comments:

Reviewer's Responses to Questions

**Comments to the Author**

1. If the authors have adequately addressed your comments raised in a previous round of review and you feel that this manuscript is now acceptable for publication, you may indicate that here to bypass the “Comments to the Author” section, enter your conflict of interest statement in the “Confidential to Editor” section, and submit your "Accept" recommendation.

Reviewer #2: All comments have been addressed

2. Is the manuscript technically sound, and do the data support the conclusions?

Reviewer #2: Yes

3. Has the statistical analysis been performed appropriately and rigorously? 

Reviewer #2: I Don't Know

4. Have the authors made all data underlying the findings in their manuscript fully available?

Reviewer #2: Yes

5. Is the manuscript presented in an intelligible fashion and written in standard English?

Reviewer #2: Yes

6. Review Comments to the Author

Reviewer #2: (No Response)

7. PLOS authors have the option to publish the peer review history of their article (what does this mean?). If published, this will include your full peer review and any attached files.

Reviewer #2: **Yes: **Hazel Keedle

---

## [Editor Report · Acceptance letter]

4 Jul 2022

PONE-D-22-04372R1 

A direct comparison of patient-reported outcomes and experiences in alternative models of maternity care in Queensland, Australia 

Dear Dr. Miller:

I'm pleased to inform you that your manuscript has been deemed suitable for publication in PLOS ONE. Congratulations! Your manuscript is now with our production department. 

Kind regards, 

on behalf of

Dr. Hannah Dahlen 

Academic Editor

PLOS ONE